# COMMUNICATION-COMPUTATION EFFICIENT SECURE AGGREGATION FOR FEDERATED LEARNING

## ABSTRACT

Federated learning has been spotlighted as a way to train neural network models using data distributed over multiple clients without a need to share private data. Unfortunately, however, it has been shown that data privacy could not be fully guaranteed as adversaries may be able to extract certain information on local data from the model parameters transmitted during federated learning. A recent solution based on the secure aggregation primitive enables privacy-preserving federated learning, but at the expense of significant extra communication/computational resources. In this paper, we propose communication-computation efficient secure aggregation which reduces the amount of communication/computational resources at least by a factor of $\sqrt{n/\log n}$ relative to the existing secure solution without sacrificing data privacy, where $n$ is the number of clients. The key idea behind the suggested scheme is to design the topology of the secret-sharing nodes (denoted by the assignment graph $G$) as sparse random graphs instead of the complete graph corresponding to the existing solution. We first obtain a sufficient condition on $G$ to guarantee reliable and private federated learning. Afterwards, we suggest using the Erdős-Rényi graph as $G$, and provide theoretical guarantees on the reliability/privacy of the proposed scheme. Through extensive real-world experiments, we demonstrate that our scheme, using only $50\%$ of the resources required in the conventional scheme, maintains virtually the same levels of reliability and data privacy in practical federated learning systems.

## 1 INTRODUCTION

Federated learning (McMahan et al., 2017) has been considered as a promising framework for training models in a decentralized manner without explicitly sharing the local private data. This framework is especially useful in various predictive models which learn from private distributed data, *e.g.*, healthcare services based on medical data distributed over multiple organizations (Brisimi et al., 2018; Xu & Wang, 2019) and text prediction based on the messages of distributed clients (Yang et al., 2018; Ramaswamy et al., 2019). In the federated learning (FL) setup, each device contributes to the global model update by transmitting its local model only; the private data is not shared across the network, which makes FL highly attractive (Kairouz et al., 2019; Yang et al., 2019).

Unfortunately, however, FL could still be vulnerable against the adversarial attacks on the data leakage. Specifically, the local model transmitted from a device contains extensive information on the training data, and an eavesdropper can estimate the data owned by the target device (Fredrikson et al., 2015; Shokri et al., 2017; Melis et al., 2019). Motivated by this issue, the authors of (Bonawitz et al., 2017) suggested *secure aggregation* (SA), which integrates cryptographic primitives into the FL framework to protect data privacy. However, SA requires significant amounts of additional resources on communication and computing for guaranteeing privacy. Especially, the communication and computation burden of SA increases as a quadratic function of the number of clients, which limits the scalability of SA.

**Contributions** We propose communication-computation efficient secure aggregation (CCESA), which maintains the reliability and data privacy in federated learning, with reduced resources on communication and computation compared to conventional SA. Our basic idea is illustrated in Fig. 1

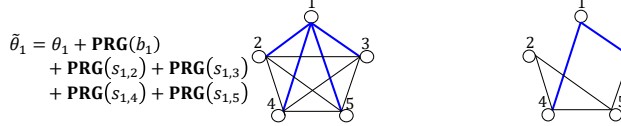

(a) Existing algorithm      (b) Suggested algorithm (CCESA)

Figure 1: Conventional secure aggregation (SA) (Bonawitz et al., 2017) versus the suggested communication-computation efficient secure aggregation (CCESA). Via selective secret sharing across only a subset of client pairs, the proposed algorithm reduces the communication cost (for exchanging public keys and secret shares among clients) and computational cost (for generating secret shares and pseudo-random values, and performing key agreements), compared to the existing fully-shared method. CCESA still maintains virtually the same levels of reliability and privacy, as proven by the theoretic analysis of Section 4.

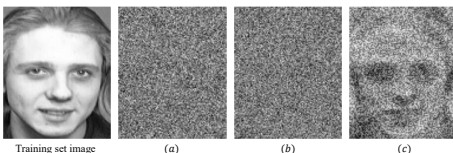

Figure 2: A training image and the reconstructed images using model inversion attacks with $(a)$ the proposed scheme (CCESA), $(b)$ existing secure aggregation (SA) (Bonawitz et al., 2017), $(c)$ federated averaging with no security measures (McMahan et al., 2017). The federated averaging scheme leaks private data from the transmitted model, while SA and the proposed CCESA do not. Note that the required communication/computational resources of CCESA are only 40% of those of SA. Additional examples are given in Supplementary Materials.

for $n = 5$ clients. Compared to the existing scheme (Bonawitz et al., 2017), which applies secret sharing for all client pairs, we suggest sharing secrets for a subset of pairs in a way that data privacy is preserved. Using theoretic analysis, we provide a sufficient condition on the graph topology for private and reliable federated learning. As summarized in Table 1, the proposed CCESA algorithm maintains both reliability and privacy against an eavesdropper who can access to any information transmitted between every client and the server, while the required amount of resources reduces by a factor of at least $O(\sqrt{n/\log n})$ compared to SA. Notably, the reduction factor gets bigger with increasing $n$, suggesting that the proposed scheme is a scalable solution for privacy-preserving federated learning. Our mathematical results are also confirmed in experiments on two real datasets of AT&T face database and CIFAR-10. Especially, under the model inversion attack on the face dataset, the results in Fig. 2 show that the suggested scheme achieves perfect data privacy by using less amount of resources than SA, while federated averaging without security measures (McMahan et al., 2017) significantly compromises the privacy of the data.

**Related work** Focusing on the collaborative learning setup with multiple clients and a server, previous works have suggested solutions to prevent the information leakage in the communication links between the server and clients. One major approach utilizes the concept of differential privacy (DP) (Dwork et al., 2014) by adding artificial random noise to the transmitted models (Wei et al., 2020; Geyer et al., 2017; Truex et al., 2020) or gradients (Shokri & Shmatikov, 2015; Abadi et al., 2016; Balcan et al., 2012). Depending on the noise distribution, DP-based collaborative learning generally exhibits a trade-off between the privacy level and the convergence of the global model.

Another popular approach is deploying secure multiparty computation (MPC) (Ben-Or et al., 1988; Damgård et al., 2012; Aono et al., 2017; Lindell et al., 2015; Bonawitz et al., 2017; Zhang et al., 2018; Tjell & Wisniewski, 2019; Shen et al., 2020; So et al., 2020) based on the cryptographic primitives including secret sharing and the homomorphic encryption (Leontiadis et al., 2014; 2015; Shi et al., 2011; Halevi et al., 2011). Although these schemes guarantee privacy, they suffer from high communication burdens while reconstructing the secret information distributed over multiple clients. A notable work (Bonawitz et al., 2017) suggested *secure aggregation* (SA), which tolerates multiple client failures by applying pairwise additive masking (Ács & Castelluccia, 2011; Elahi et al., 2014; Jansen & Johnson, 2016; Goryczka & Xiong, 2015). A recent work (So et al., 2020) suggested Turbo-aggregate, which partitions $n$ computing devices into $L$ groups and updates the global model by utilizing the circular aggregation topology. However, each client in Turbo-aggregate requires a communication cost of at least $4mnR/L$ bits, which is much larger than that of our scheme (CCESA) requiring a communication cost of $\sqrt{n \log n}(2a_K + 5a_S) + mR$[1]. For example,

---

[1]Here, $m$ is the number of model parameters where each parameter is represented in $R$ bits. $a_K$ and $a_S$ are the number of bits required for exchanging public keys and the number of bits in a secret share, respectively.

|  |  | CCESA | SA | FedAvg |
|---|---|---|---|---|
| Communication cost | Client | $O(\sqrt{n\log n}+m)$ | $O(n+m)$ | $O(m)$ |
|  | Server | $O(n\sqrt{n\log n}+mn)$ | $O(n^2+mn)$ | $O(mn)$ |
| Computation cost | Client | $O(n\log n+m\sqrt{n\log n})$ | $O(n^2+mn)$ | $0$ |
|  | Server | $O(mn\log n)$ | $O(mn^2)$ | $O(mn)$ |
| Reliability | | $\geq 1-O(ne^{-\sqrt{n\log n}})$ | $\geq 1-O(ne^{-n})$ | $1$ |
| Privacy | | $\geq 1-O(n^{-C})$, for $C>0$ | $1$ | $0$ |

Table 1: Communication and computation cost of the proposed CCESA algorithm, Secure Aggregation (SA) in (Bonawitz et al., 2017), federated averaging (FedAvg) in (McMahan et al., 2017). Detailed expressions and derivations are in Section 4 and Supplementary Materials, respectively.

in a practical scenario with $m=10^6, R=32, n=100, L=10$ and $a_K = a_S = 256$, our scheme requires only 3% of the communication bandwidth used in Turbo-aggregate.

The idea of replacing the complete graph with a low-degree graph for communication efficiency has been studied in the areas of distributed learning (Charles et al., 2017; Sohn et al., 2020) and multi-party computation (Fitzi et al., 2007; Harnik et al., 2007). A very recent paper (Bell et al., 2020) proposed new protocols using $k$-regular graphs for communication in the secure aggregation (Bonawitz et al., 2017) framework, which are robust against semi-honest and semi-malicious threat models, respectively. However, the results in (Bell et al., 2020) is based on a strong assumption on the number of clients dropped out of the protocol, while our work does not assume anything on the number of dropouts.

## 2 BACKGROUND

**Federated learning**  Consider a scenario with one server and $n$ clients. Each client $i$ has its local training dataset $\mathcal{D}_i = \{(\mathbf{x}_{i,k}, y_{i,k})\}_{k=1}^{N_i}$ where $\mathbf{x}_{i,k}$ and $y_{i,k}$ are the feature vector and the label of the $k$-th training sample, respectively. For each round $t$, the server first selects a set $S_t$ of $cn$ clients $(0 < c \leq 1)$ and sends the current global model $\theta_{\text{global}}^{(t)}$ to those selected clients. Then, each client $i \in S_t$ updates the received model by using the local data $\mathcal{D}_i$ and sends the updated model $\theta_i^{(t+1)}$ to the server. Finally, the server updates the global model by aggregating local updates from the selected clients, *i.e.*, $\theta_{\text{global}}^{(t+1)} \leftarrow \sum_{i \in S_t} \frac{N_i}{N} \theta_i^{(t+1)}$ where $N = \sum_{i \in S_t} N_i$.

**Cryptographic primitives for preserving privacy**  Here we review three cryptographic tools used in SA (Bonawitz et al., 2017). First, *t-out-of-n secret sharing* (Shamir, 1979) is splitting a secret $s$ into $n$ shares, in a way that any $t$ shares can reconstruct $s$, while any $t-1$ shares provides absolutely no information on $s$. We denote $t$-out-of-$n$ secret sharing by $s \xrightarrow{(t,n)} (s_k)_{k \in [n]}$, where $s_k$ indicates the $k^{\text{th}}$ share of secret $s$ and $[n]$ represents the index set $\{1, 2, \cdots, n\}$. Second, the *Diffie-Hellman key agreement* is used to generate a secret $s_{i,j}$ that is only shared by two target clients $i, j \in [n]$. The key agreement scheme designs public-private key pairs $(s_u^{PK}, s_u^{SK})$ for clients $u \in [n]$ in a way that $s_{i,j} = f(s_i^{PK}, s_j^{SK}) = f(s_j^{PK}, s_i^{SK})$ holds for all $i, j \in [n]$ for some key agreement function $f$. The secret $s_{i,j}$ is unknown when neither $s_i^{SK}$ nor $s_j^{SK}$ is provided. Third, *symmetric authenticated encryption* is encrypting/decrypting message $m$ using a key $k$ shared by two target clients. This guarantees the integrity of the messages communicated by the two clients.

**Secure aggregation (Bonawitz et al., 2017)**  For privacy-preserving federated learning, SA has been proposed based on the cryptographic primitives of Shamir's secret sharing, key agreement and symmetric authenticated encryption. The protocol consists of four steps: **Step 0** Advertise Keys, **Step 1** Share Keys, **Step 2** Masked Input Collection, and **Step 3** Unmasking.

Consider a server with $n$ clients where client $i \in [n]$ has its private local model $\theta_i$. Denote the client index set as $V_0 = V = [n]$. The objective of the server is to obtain the sum of models $\sum_i \theta_i$ without getting any other information on private local models. In **Step 0**, client $i \in V_0$ generates key pairs $(s_i^{PK}, s_i^{SK})$ and $(c_i^{PK}, c_i^{SK})$ by using a key agreement scheme. Then, client $i$ advertises its public keys $(s_i^{PK}, c_i^{PK})$ to the server. The server collects the public keys from a client set $V_1 \subset V_0$, and broadcasts $\{(i, s_i^{PK}, c_i^{PK})\}_{i \in V_1}$ to all clients in $V_1$. In **Step 1**, client $i$ generates a random element $b_i$ and applies $t$-out-of-$n$ secret sharing to generate $n$ shares of $b_i$ and $s_i^{SK}$, *i.e.*, $b_i \xrightarrow{(t,n)} (b_{i,j})_{j \in [n]}$ and $s_i^{SK} \xrightarrow{(t,n)} (s_{i,j}^{SK})_{j \in [n]}$. By using the symmetric authenticated encryption, client $i$ computes the

ciphertext $e_{i,j}$ for all $j \in V_1 \backslash \{i\}$, by taking $b_{i,j}$ and $s_{i,j}^{SK}$ as messages and $c_{i,j} = f(c_j^{PK}, c_i^{SK})$ as a key, where $f$ is the key agreement function. Finally, client $i$ sends $\{(i, j, e_{i,j})\}_{j \in V_1 \backslash \{i\}}$ to the server. The server collects the message from at least $t$ clients (denote this set of client as $V_2 \subset V_1$) and sends $\{(i, j, e_{i,j})\}_{i \in V_2}$ to each client $j \in V_2$. In **Step 2**, client $i$ computes the shared secret $s_{i,j} = f(s_j^{PK}, s_i^{SK})$ for all $j \in V_2 \backslash \{i\}$. Then, client $i$ computes the masked private vector

$$\tilde{\theta}_i = \theta_i + \mathbf{PRG}(b_i) + \sum_{j \in V_2; i < j} \mathbf{PRG}(s_{i,j}) - \sum_{j \in V_2; i > j} \mathbf{PRG}(s_{i,j}), \tag{1}$$

and sends $\tilde{\theta}_i$ to the server, where $\mathbf{PRG}(x)$ indicates a pseudorandom generator with seed $x$ outputting a vector having the dimension identical to $\theta_i$. Note that the masked vector $\tilde{\theta}_i$ gives no information on private vector $\theta_i$ unless both $s_i^{SK}$ and $b_i$ are revealed. The server collects $\tilde{\theta}_i$ from at least $t$ clients (denote this set as $V_3 \subset V_2$), and sends $V_3$ to each client $i \in V_3$. In **Step 3**, client $j$ decrypts the ciphertext $\{e_{i,j}\}_{i \in V_2 \backslash \{j\}}$ by using the key $c_{i,j} = f(c_i^{PK}, c_j^{SK})$ to obtain $\{b_{i,j}\}_{i \in V_2 \backslash \{j\}}$ and $\{s_{i,j}^{SK}\}_{i \in V_2 \backslash \{j\}}$. Each client $j$ sends a set of shares $\{b_{i,j}\}_{i \in V_3} \cup \{s_{i,j}^{SK}\}_{i \in V_2 \backslash V_3}$ to the server. The server collects the responds from at least $t$ clients (denote this set of clients as $V_4 \subset V_3$). For each client $i \in V_3$, the server reconstructs $b_i$ from $\{b_{i,j}\}_{i \in V_4}$ and computes $\mathbf{PRG}(b_i)$. Similarly, for each client $i \in V_2 \backslash V_3$, the server reconstructs $s_i^{SK}$ from $\{s_{i,j}^{SK}\}_{i \in V_4}$ and computes $\mathbf{PRG}(s_{i,j})$ for all $j \in V_2$. Using this information, the server obtains the sum of private local models by computing

$$\sum_{i \in V_3} \theta_i = \sum_{i \in V_3} \tilde{\theta}_i - \sum_{i \in V_3} \mathbf{PRG}(b_i) - \sum_{j \in V_2, i \in V_2 \backslash V_3; i < j} \mathbf{PRG}(s_{i,j}) + \sum_{j \in V_2, i \in V_2 \backslash V_3; i > j} \mathbf{PRG}(s_{i,j}). \tag{2}$$

## 3 SUGGESTED ALGORITHM

In the secure aggregation (Bonawitz et al., 2017), the public keys $(c_i^{PK}, s_i^{PK})$ and the shares of secrets $(s_i^{SK}, b_i)$ are transmitted between clients and the server, which requires additional communication/computational resources compared with the vanilla federated learning. Specifically, since each client $i$ needs to receive information from all other clients $j \neq i$, the required amount of resources increases as a quadratic function of the number of clients $n$.

In this paper, we suggest a variant of secure aggregation, dubbed as communication-computation efficient secure aggregation (CCESA), which enables to provide a more scalable solution for privacy-preserving federated learning by improving the communication/computational efficiency. The basic idea of the proposed algorithm is to allow each client to share its public keys and secret shares to a *subset* of other clients, instead of sharing them with all other clients. By doing so, compared with SA, the suggested scheme achieves two advantages in resource efficiency, without losing the reliability of learning algorithms and the data privacy. The first advantage is the reduction of the communication cost, since each node shares its public keys and secrets with less clients. The second advantage is a reduction of the computational cost of each client, since a smaller number of masks are used while computing its masked private vector.

The proposed algorithm is specified by the *assignment graph* which represents how public keys and secret shares are assigned to the other clients. Given $n$ clients, the assignment graph $G = (V, E)$ consists of $n$ vertices where the vertex and the edge set of $G$ are represented by $V$ and $E$, respectively. We set $V = [n]$ where each index $i \in V$ represents client $i$, and the edge $\{i, j\} \in E$ connecting vertices $i$ and $j$ indicates that client $i$ and $j$ exchange their public keys and secret shares. For vertex $i \in [n]$, we define $Adj(i) := \{j; \{i, j\} \in E\}$ as the index set of vertices adjacent to vertex $i$. In our algorithm, public keys and secrets of client $i$ are shared with clients $j \in Adj(i)$.

Now, using the assignment graph notation, we formally define the suggested algorithm. Due to the space limitation, we put the algorithm in Supplementary Materials A; here we only describe what differs from SA. In **Step 0**, instead of broadcasting the public keys $(c_j^{PK}, s_j^{PK})$ for client $j$ to all other clients, the server sends the public keys only to the client $i$ satisfying $j \in Adj(i) \cap V_1$. In **Step 1**, each client $i \in V_1$ uses $t_i$-out-of-$(|Adj(i)| + 1)$ secret sharing scheme to generate shares of $s_i^{SK}$ and $b_i$, i.e., $s_i^{SK} \xrightarrow{(t_i, |Adj(i)|+1)} (s_{i,j}^{SK})_{j \in Adj(i) \cup \{i\}}$ and $b_i \xrightarrow{(t_i, |Adj(i)|+1)} (b_{i,j})_{j \in Adj(i) \cup \{i\}}$, and sends the encrypted $s_{i,j}^{SK}$ and $b_{i,j}$ to client $j$ through the server. In **Step 2**, client $i$ computes the masked private model

$$\tilde{\theta}_i = \theta_i + \mathbf{PRG}(b_i) + \sum_{j \in V_2 \cap Adj(i); i < j} \mathbf{PRG}(s_{i,j}) - \sum_{j \in V_2 \cap Adj(i); i > j} \mathbf{PRG}(s_{i,j}), \tag{3}$$

and transmits $\tilde{\theta}_i$ to the server. In **Step 3**, client $i$ sends $b_{j,i}$ to the server for all $j \in V_3 \cap Adj(i)$, and sends $s_{j,i}^{SK}$ to the server for all $j \in (V_2 \backslash V_3) \cap Adj(i)$. After reconstructing secrets from shares, the server obtains the sum of the local models $\theta_i$ as

$$\sum_{i \in V_3} \theta_i = \sum_{i \in V_3} \tilde{\theta}_i - \sum_{i \in V_3} \mathbf{PRG}(b_i) - \sum_{i \in V_2 \backslash V_3, j \in Adj(i) \cap V_3; i>j} \mathbf{PRG}(s_{i,j}) + \sum_{i \in V_2 \backslash V_3, j \in Adj(i) \cap V_3; i<j} \mathbf{PRG}(s_{i,j}). \tag{4}$$

Note that the suggested protocol with $n$-complete assignment graph $G$ reduces to SA.

Here we define several notations representing the evolution of the assignment graph $G$ as some of the nodes may drop out of the system in each step. Recall that $V_0 = V$ and $V_{i+1}$ is defined as the set of survived nodes in Step $i \in \{0, \cdots, 3\}$. Let us define $G_i$ as the induced subgraph of $G$ whose vertex set is $V_i$, i.e., $G_i := G - (V \backslash V_i)$. Then, $G_{i+1}$ represents how the nodes survived until Step $i$ are connected. We define the evolution of assignment graph during the protocol as $\boldsymbol{G} = (G_0, G_1, \cdots, G_4)$.

In Fig. 1, we illustrate an example of the suggested algorithm with $n = 5$ clients. Fig. 1a corresponds to SA (Bonawitz et al., 2017), while Fig. 1b depicts the proposed scheme. Here, we focus on the required communication/computational resources of client 1. Note that each client exchanges public keys and secret shares with its adjacent clients. For example, client 1 exchanges the data with four other clients in the conventional scheme, while client 1 exchanges the data with clients 3 and 5 in the suggested scheme. Thus, the proposed CCESA requires only half of the bandwidth compared to the conventional scheme. In addition, CCESA requires less computational resources than conventional scheme, since each client generates less secret shares and pseudo-random values, and performs less key agreements.

## 4 THEORETICAL ANALYSIS

### 4.1 PERFORMANCE METRICS

The proposed CCESA algorithm aims at developing private, reliable and resource-efficient solutions for federated learning. Here, we define key performance metrics for federated learning systems including federated averaging (McMahan et al., 2017), SA (Bonawitz et al., 2017) and CCESA. Recall that the server receives (masked) model parameters $\tilde{\theta}_i$ from clients $i \in V_3$, and wants to update the global model as the sum of the unmasked model parameters, i.e., $\theta_{\text{global}} \leftarrow \sum_{i \in V_3} \theta_i$. The condition for successful (or reliable) global model update is stated as follows.

**Definition 1.** *A system is called reliable if the server successfully obtains the sum of the model parameters $\sum_{i \in V_3} \theta_i$ aggregated over the distributed clients.*

Now, we define private federated learning. In our analysis, we focus on a passive eavesdropper who can access to any information transmitted between any client and the server throughout running CCESA algorithm, namely, public keys of clients, secret shares, masked local models and the indices of survived clients $V_3$[2]. Once an eavesdropper gets the local model $\theta_i$ of client $i$, it can reconstruct the private data (*e.g.*, face image or medical records) of the client or can identify the client which contains the target data. In general, if an eavesdropper obtains the sum of local models occupied by a subset $\mathcal{T}$ of clients, a similar privacy attack is possible for the subset of clients. Thus, to preserve the data privacy, it is safe to protect the information on the partial sum of model parameters against the eavesdropper; we formalize this.

**Definition 2.** *A system is called private if $H(\sum_{i \in \mathcal{T}} \theta_i) = H(\sum_{i \in \mathcal{T}} \theta_i | E)$ holds for all $\mathcal{T}$ satisfying $\mathcal{T} \subset V_3$ and $\mathcal{T} \notin \{\varnothing, V_3\}$. Here, $H$ is the entropy function, and $E$ is the information accessible to the eavesdropper.*

When both reliability and privacy conditions hold, the server successfully updates the global model, while an eavesdropper cannot extract data in the information-theoretic sense. We define $P_e^{(r)}$ and $P_e^{(p)}$ as the probabilities that reliability and privacy conditions do not hold, respectively.

---

[2]Our threat model is equivalent to the "server-only" honest-but-curious adversary, which is weaker than "client-server collusion" adversary considered in secure aggregation (Bonawitz et al., 2017).

### 4.2 RESULTS FOR GENERAL ASSIGNMENT GRAPH $G$

Recall that our proposed scheme is specified by the assignment graph $G$. We here provide mathematical analysis on the performance metrics of reliability and privacy, in terms of the graph $G$. To be specific, the theorems below provide the necessary and sufficient conditions on the assignment graph $G$ to enable reliable/private federated learning, where the reliability and privacy are defined in Definitions 1 and 2. Before going into the details, we first define *informative* nodes as below.

**Definition 3.** *A node $i \in V_0$ is informative if $|(Adj(i) \cup \{i\}) \cap V_4| \geq t_i$ holds.*

Note that node $i$ is called informative when the server can reconstruct the secrets ($b_i$ or $s_i^{SK}$) of node $i$ in Step 3 of the algorithm. Using this definition, we state the condition on graph $G$ for enabling reliable systems as below.

**Theorem 1.** *The system is reliable if and only if node $i$ in informative for all $i \in V_3^+$, where $V_3^+ = V_3 \cup \{i \in V_2 : Adj(i) \cap V_3 \neq \varnothing\}$ is the union of $V_3$ and the neighborhoods of $V_3$ within $V_2$.*

*Proof.* The full proof is given in Supplementary Materials; here we provide a sketch for the proof. Recall that the server receives the sum of masked models $\sum_{i \in V_3} \tilde{\theta}_i$, while the system is said to be reliable if the server obtains the sum of unmasked models $\sum_{i \in V_3} \theta_i$. Thus, the reliability condition holds if and only if the server can cancel out the random terms in (4), which is possible when either $s_i^{SK}$ or $b_i$ is recovered for all $i \in V_3^+$. Since a secret is recovered if and only if at least $t_i$ shares are gathered from adjacent nodes, we need $|(Adj(i) \cup \{i\}) \cap V_4| \geq t_i$, which completes the proof. $\square$

Now, before moving on to the next theorem, we define some sets of graph evolutions as below:

$$\mathcal{G}_{\mathrm{C}} = \{\boldsymbol{G} = (G_0, G_1, \cdots, G_4) : G_3 \text{ is connected }\},$$
$$\mathcal{G}_{\mathrm{D}} = \{\boldsymbol{G} = (G_0, G_1, \cdots, G_4) : G_3 \text{ is not connected }\},$$
$$\mathcal{G}_{\mathrm{NI}} = \{\boldsymbol{G} \in \mathcal{G}_{\mathrm{D}} : \forall l \in [\kappa], \exists i \in C_l^+ \text{ such that node } i \text{ is not informative}\}.$$

Here, when $G_3$ is a disconnected graph with $\kappa \geq 2$ components, $C_l$ is defined as the vertex set of the $l^{\text{th}}$ component, and $C_l^+ := C_l \cup \{i \in V_2 : Adj(i) \cap C_l \neq \varnothing\}$. Using this definition, we state a sufficient condition on the assignment graph to enable private federated learning.

**Lemma 1.** *The system is private if $\boldsymbol{G} \in \mathcal{G}_C$.*

*Proof.* Again we just provide a sketch of the proof here; the full proof is in Supplementary Materials. Note that $G_3$ is the induced subgraph of $G$ whose vertex set is $V_3$. Suppose an eavesdropper has access to the masked local models $\{\tilde{\theta}_i\}_{i \in \mathcal{T}}$ of a subset $\mathcal{T} \subset V_3$ of nodes. Now, the question is whether this eavesdropper can recover the sum of the unmasked models $\sum_{i \in \mathcal{T}} \theta_i$. If $G_3$ is connected, there exists an edge $e = \{p, q\}$ such that $p \in \mathcal{T}$ and $q \in V_3 \backslash \mathcal{T}$. Note that $\sum_{i \in \mathcal{T}} \tilde{\theta}_i$ contains the $\mathbf{PRG}(s_{p,q})$ term, while $s_{p,q}$ is not accessible by the eavesdropper since $p, q \in V_3$. Thus, from (4), the eavesdropper cannot obtain $\sum_{i \in \mathcal{T}} \theta_i$, which completes the proof. $\square$

Based on the Lemma above, we state the necessary and sufficient condition for private system as below, the proof of which is given in the Supplementary Materials.

**Theorem 2.** *The system is private if and only if $\boldsymbol{G} \in \mathcal{G}_C \cup \mathcal{G}_{NI}$.*

The theorems above provide guidelines on how to construct the assignment graph $G$ to enable reliable and private federated learning. These guidelines can be further specified when we use the Erdős-Rényi graph as the assignment graph $G$. In the next section, we explore how the Erdős-Rényi graph can be used for reliable and private federated learning.

### 4.3 RESULTS FOR ERDŐS-RÉNYI ASSIGNMENT GRAPH $G$

The Erdős-Rényi graph $G \in G(n, p)$ is a random graph of $n$ nodes where each edge connecting two arbitrary nodes is connected with probability $p$. Define CCESA$(n, p)$ as the proposed scheme using the assignment graph of $G \in G(n, p)$. According to the analysis provided in this section, CCESA$(n, p)$ almost surely achieves both reliability and privacy conditions, provided that the connection probability $p$ is chosen appropriately. Throughout the analysis below, we assume that each client independently drops out with probability $q$ at each step (from Step 0 to Step 3), and the secret sharing parameter $t_i$ is set to $t$ for all $i \in [n]$.

### 4.3.1 FOR ASYMPTOTICALLY LARGE $n$

We start with the analysis on CCESA$(n, p)$ when $n$ is asymptotically large. The following two theorems provide lower bounds on $p$ to satisfy reliability/privacy conditions. The proofs are provided in Supplementary Materials.

**Theorem 3.** *CCESA$(n, p)$ is asymptotically almost surely reliable if $p > \frac{3\sqrt{(n-1)\log(n-1)} - 1}{(n-1)(2(1-q)^4 - 1)}$.*

**Theorem 4.** *CCESA$(n, p)$ is asymptotically almost surely private if $p > \frac{\log(\lceil n(1-q)^3 - \sqrt{n\log n}\rceil)}{\lceil n(1-q)^3 - \sqrt{n\log n}\rceil}$.*

From these theorems, the condition for achieving both reliability and privacy is obtained as follows.

**Remark 1.** *Let*

$$p^\star = \max\{\frac{\log(\lceil n(1-q)^3 - \sqrt{n\log n}\rceil)}{\lceil n(1-q)^3 - \sqrt{n\log n}\rceil}, \frac{3\sqrt{(n-1)\log(n-1)} - 1}{(n-1)(2(1-q)^4 - 1)}\}. \tag{5}$$

*If $p > p^\star$, then CCESA$(n, p)$ is asymptotically almost surely (a.a.s.) reliable and private. Note that the threshold connection probability $p^\star$ is a decreasing function of $n$. Thus, the proposed algorithm is getting more resource efficient than SA as $n$ grows, improving the scalability of the system.*

In the remarks below, we compare SA and the proposed CCESA, in terms of the required amount of communication/computational resources to achieve both reliability and privacy. These results are summarized in Table 1.

**Remark 2.** *Let $B$ be the amount of additional communication bandwidth used at each client, compared to that of federated averaging (McMahan et al., 2017). Since the bandwidth is proportional to $np$, we have $B_{CCESA(n,p)} \sim O(\sqrt{n\log n})$ and $B_{SA} \sim O(n)$. Thus, the suggested CCESA protocol utilizes a much smaller bandwidth compared to SA in (Bonawitz et al., 2017). The detailed comparison is given in Section D.1 of the Supplementary Materials.*

**Remark 3.** *Compared to SA, the proposed CCESA algorithm generates a smaller number of secret shares and pseudo-random values, and performs less key agreements. Thus, the computational burden at the server and the clients reduces by a factor of at least $O(\sqrt{n/\log n})$. The detailed comparison is given in Section D.2 of the Supplementary Materials.*

### 4.3.2 FOR FINITE $n$

We now discuss the performance of the suggested scheme for finite $n$. Let $P_e^{(p)}$ be the error probability that CCESA$(n, p)$ does not satisfy the the privacy condition, and define $P_e^{(p)}$ as the error probability that CCESA$(n, p)$ is not reliable. Below we provide upper bounds on $P_e^{(p)}$ and $P_e^{(r)}$.

**Theorem 5.** *For arbitrary $n, p, q$ and $t$, the error probability for reliability $P_e^{(r)}$ is bounded by $P_e^{(r)} \leq ne^{-(n-1)D_{KL}(\frac{t-1}{n-1}||p(1-q)^4)}$, where $D_{KL}$ is the Kullback-Leibler (KL) divergence.*

**Theorem 6.** *For arbitrary $n, p$ and $q$, the error probability for privacy $P_e^{(p)}$ is bounded by*

$$P_e^{(p)} \leq \sum_{m=0}^{n} \binom{n}{m}(1-q)^{3m}(1-(1-q)^3)^{(n-m)} \sum_{k=1}^{\lfloor m/2\rfloor} \binom{m}{k}(1-p)^{k(m-k)}.$$

Fig. 3 illustrates the upper bounds on $P_e^{(p)}$ and $P_e^{(r)}$ obtained in Theorems 5 and 6, when $p = p^\star$. Here, $q_{\text{total}} := 1 - (1-q)^4$ is defined as the dropout probability of the entire protocol (from Step 0 to Step 3). Note that the upper bounds in Theorems 5 and 6 are decreasing functions of $p$. Therefore, the plotted values in Fig. 3 are indeed upper bounds on the error probabilities for arbitrary $p > p^\star$. It is shown that a system with the suggested algorithm is private and reliable with high probability for an arbitrary chosen $p > p^\star$. The error probability for the privacy $P_e^{(p)}$ is below $10^{-40}$, which is negligible even for small $n$. The error probability for the reliability $P_e^{(r)}$ is below $10^{-2}$, which means that in at most one round out of 100 federated learning rounds, the (masked) models $\{\tilde{\theta}_i\}_{i \in V_3}$ received by the server cannot be converted to the sum of (unmasked) local models $\sum_{i \in V_3} \theta_i$. Even in this round when the server cannot obtain the sum of (unmasked) local models, the server is aware of the fact that the current round is not reliable, and may maintain the global model used in the previous round. This does not harm the accuracy of our scheme, as shown in the experimental results of Section 5.

Figure 3: Upper bounds on the error probabilities $P_e^{(r)}$ and $P_e^{(p)}$ in Theorems 5 and 6 for $p = p^\star$, where $p^\star$ is the threshold connection probability for achieving both reliability and privacy as in (5). Note that for arbitrary $p > p^\star$, the error probabilities are lower than the upper bounds marked in the figure. One can confirm that the suggested CCESA algorithm is private and reliable with a high probability, provided that $p > p^\star$.

| | $n$ | $q_{\text{total}}$ | $t$ | $p$ | Client | | | | Server |
|---|---|---|---|---|---|---|---|---|---|
| | | | | | Step 0 | Step 1 | Step 2 | Step 3 | |
| SA | 100 | 0 | 51 | 1 | 6 | 3572 | 3537 | 88 | 13 |
| | 100 | 0.1 | 51 | 1 | 6 | 3540 | 3365 | 80 | 9847 |
| | 300 | 0 | 151 | 1 | 6 | 11044 | 10867 | 269 | 82 |
| | 300 | 0.1 | 151 | 1 | 6 | 10502 | 10453 | 255 | 72847 |
| | 500 | 0 | 251 | 1 | 6 | 19097 | 18196 | 449 | 198 |
| | 500 | 0.1 | 251 | 1 | 6 | 18107 | 17315 | 432 | 329645 |
| CCESA | 100 | 0 | 43 | 0.6362 | 6 | 2216 | 2171 | 56 | 16 |
| | 100 | 0.1 | 51 | 0.7953 | 6 | 2715 | 2648 | 66 | 8067 |
| | 300 | 0 | 83 | 0.4109 | 6 | 4435 | 4354 | 110 | 54 |
| | 300 | 0.1 | 98 | 0.5136 | 6 | 5382 | 5300 | 113 | 37082 |
| | 500 | 0 | 112 | 0.3327 | 6 | 5954 | 5846 | 145 | 132 |
| | 500 | 0.1 | 133 | 0.4159 | 6 | 7634 | 7403 | 184 | 141511 |

Table 2: Running time (unit: ms) of SA (Bonawitz et al., 2017) and suggested CCESA

# 5 EXPERIMENTS

Here we provide experimental results on the proposed CCESA algorithm. We compare CCESA and secure aggregation (SA) of (Bonawitz et al., 2017) in terms of time complexity (running time), reliability, and privacy. We tested both schemes on two real datasets, AT&T Laboratories Cambridge database of faces (https://www.kaggle.com/kasikrit/att-database-of-faces) and CIFAR-10. For the AT&T face dataset containing images of 40 individuals, we considered a federated learning setup where each of $n = 40$ clients uses its own images for local training. All algorithms are implemented in python and PyTorch (Paszke et al., 2017). Codes will be made available to the public.

## 5.1 RUNNING TIME

In Table 2, we tested the running time of our CCESA and existing SA for various $n$ and $q_{\text{total}}$. Similar to the setup used in (Bonawitz et al., 2017), we assumed that each node has its local model $\boldsymbol{\theta}$ with dimension $m = 10000$, where each element of the model is chosen from the field $\mathbb{F}_{2^{16}}$. Here, $t$ is selected by following the guideline in Supplementary Materials, and $p$ is chosen as $p^\star \sim O(\sqrt{\log n/n})$ defined in (5) which is proven to meet both reliability and privacy conditions. For every $n, q_{\text{total}}$ setup, the proposed CCESA$(n, p)$ requires $p$ times less running time compared with the conventional SA of (Bonawitz et al., 2017). This is because each client generates $p$ times less number of secret shares and pseudo-random values, and performs $p$ times less number of key agreements. This result is consistent with our analysis on the computational complexity in Table 1.

## 5.2 RELIABILITY

Recall that a system is reliable if the server obtains the sum of the local models $\sum_{i \in V_3} \theta_i$. Fig. 4 shows the reliability of CCESA in CIFAR-10 dataset. We plotted the test accuracies of SA and the suggested CCESA$(n, p)$ for various $p$. Here, we included the result when $p = p^\star = 0.3106$, where $p^\star$ is the provably minimum connection probability for achieving both the reliability and privacy according to Remark 1. One can confirm that CCESA with $p = p^\star$ achieves the performance of SA in both i.i.d. and non-i.i.d. data settings, coinciding with our theoretical result in Theorem 3. Moreover, in both settings, selecting $p = 0.25$ is sufficient to achieve the test accuracy performance of SA when the system is trained for 200 rounds. Thus, the required communication/computational resources for guaranteeing the reliability, which is proportional to $np$, can be reduced to $50\%$ of the conventional wisdom in federated learning. Similar behaviors are observed from the experiments on AT&T Face dataset, as in Fig. B.1 of the Supplementary Materials.

## 5.3 PRIVACY

We first consider a privacy threat called model inversion attack (Fredrikson et al., 2015). The basic setup is as follows: the attacker eavesdrops the masked model $\tilde{\theta}_i$ sent from client $i$ to the server, and reconstructs the face image of a target client. Under this setting, we compared how the eavesdropped model reveals the information on the raw data for various schemes. As in Fig. 2 and Fig. B.2 in the

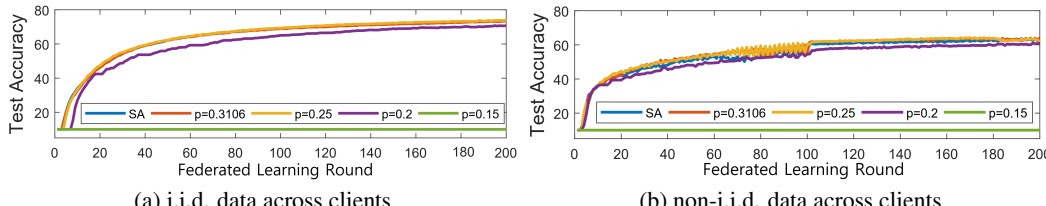

(a) i.i.d. data across clients       (b) non-i.i.d. data across clients

Figure 4: Test accuracies of SA versus proposed CCESA$(n, p)$ with various connection probability $p$, for federated learning using CIFAR-10 dataset. Here, we set $n = 1000$ and $q_{\text{total}} = 0.1$. The suggested CCESA achieves the ideal test accuracy by using only $75\%$ of the communication/computational resources used in the conventional SA. This shows the reliability of CCESA in real-world scenarios.

| Schemes \ Number of training data ($n_{\text{train}}$) | 5000 | 10000 | 15000 | 50000 |
|---|---|---|---|---|
| **Federated Averaging** (McMahan et al., 2017) | 72.49% | 70.72% | 72.80% | 66.47% |
| **Secure Aggregation (SA)** (Bonawitz et al., 2017) | 49.67% | 49.96% | 49.85% | 49.33% |
| **CCESA** (Suggested) | 49.29% | 50.14% | 49.02% | 50.00% |

Table 3: Accuracy of the membership inference attack on local models trained on CIFAR-10. The scheme with a higher attack accuracy is more vulnerable to the inference attack. In order to maximize the uncertainty of the membership inference, the test set for the attack model consists of 5000 members (training data points) and 5000 non-members (evaluation data points). For the proposed CCESA, the attacker is no better than the random guess with accuracy $= 50\%$, showing the privacy-preserving ability of CCESA.

Supplementary Materials, the vanilla federated averaging (McMahan et al., 2017) with no privacy-preserving techniques reveals the characteristics of individual's face, compromising the privacy of clients. On the other hand, both SA and CCESA do not allow any clue on the client's face; these schemes are resilient to the model inversion attack and preserve the privacy of clients. This observation is consistent with our theoretical results that the proposed CCESA guarantees data privacy. We also considered another privacy threat called the membership inference attack (Shokri et al., 2017). Here, the attacker eavesdrops the masked model $\tilde{\theta}_i$ and guesses whether a target data is a member of the training dataset. Table 3 summarizes the accuracy of the inference attack for CIFAR-10 dataset, under the federated learning setup where $n_{\text{train}}$ training data is equally distributed into $n = 10$ clients. The attack accuracy reaches near $70\%$ for federated averaging, while SA and CCESA have the attack accuracy of near $50\%$, similar to the performance of the random guess. This shows that both SA and CCESA do not reveal any clue on the local training set, which secures data privacy.

## 6 CONCLUSION

We devised communication-computation efficient secure aggregation (CCESA) which successfully preserves the data privacy of federated learning in a highly resource-efficient manner. Based on graph-theoretic analysis, we showed that using $O(n \log n)$ resources is sufficient for guaranteeing reliability and privacy of the proposed system with $n$ clients, which is much smaller than $O(n^2)$ resources used in the conventional wisdom. Our experiments on real datasets, measuring the test accuracy and the privacy leakage, show that CCESA requires only $50\%$ of resources than the conventional wisdom, to achieve the same level of reliability and privacy.

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

## A DETAILED DESCRIPTION OF THE CCESA PROTOCOL

---

**Algorithm 1:** Communication-Computation Efficient Secure Aggregation (CCESA) Protocol

**Input:** Number of clients $n$, assignment graph $G$, privacy thresholds $t_i$ of all clients $i \in [n]$, local models $\theta_i$ of all clients $i \in [n]$, Diffie-Hellman key pairs $(c_i^{PK}, c_i^{SK}), (s_i^{PK}, s_i^{SK})$ of all clients $i \in [n]$ and corresponding key agreement function $f$, pseudo-random generator **PRG**

**Step 0. Advertise Keys**

  **Client $i$:**

  Sends $(i, c_i^{PK}, s_i^{PK})$ to the server

  **Server:**

  Collects the messages from clients (denote this set of clients as $V_1$)

  Sends $\{(i, c_i^{PK}, s_i^{PK})\}_{i \in Adj(j) \cap V_1}$ to all clients $j \in V_1$;

**Step 1. Share Keys**

  **Client $i$:**

  Generates a random element $b_i$

  Applies $t_i$-out-of-$(|Adj(i)| + 1)$ secret sharing schemes to $b_i$ and $s_i^{SK}$

$$b_i \xrightarrow{(t_i, |Adj(i)|+1)} (b_{i,j})_{j \in (Adj(i)) \cup \{i\}}, \quad s_i^{SK} \xrightarrow{(t_i, |Adj(i)|+1)} (s_{i,j}^{SK})_{j \in Adj(i) \cup \{i\}}$$

  Encrypts $[b_{i,j}, s_{i,j}^{SK}]$ to $[\bar{b}_{i,j}, \bar{s}_{i,j}^{SK}]$ using the authenticated encryption with key $f(c_j^{PK}, c_i^{SK})$

  Sends $\{(i, j, \bar{b}_{i,j}, \bar{s}_{i,j}^{SK})\}_{j \in Adj(i) \cap V_1}$ to the server

  **Server:**

  Collects the messages from clients (denote this set of clients as $V_2$)

  Sends $\{(i, j, \bar{b}_{i,j}, \bar{s}_{i,j}^{SK})\}_{i \in Adj(j) \cap V_2}$ to all clients $j \in V_2$

**Step 2. Masked Input Collection**

  **Client $i$:**

  Computes $s_{i,j} = f(s_j^{PK}, s_i^{SK})$ and

$$\tilde{\theta}_i = \theta_i + \textbf{PRG}(b_i) + \sum_{j \in V_2 \cap Adj(i); i < j} \textbf{PRG}(s_{i,j}) - \sum_{j \in V_2 \cap Adj(i); i > j} \textbf{PRG}(s_{i,j})$$

  Sends $(i, \tilde{\theta}_i)$ to the server

  **Server:**

  Collects the messages from clients (denote this set of clients as $V_3$)

  Sends $V_3$ to all clients $j$ in $V_3$

**Step 3. Unmasking**

  **Client $i$:**

  Decrypts $\bar{b}_{i,j}$ with key $f(c_j^{PK}, c_i^{SK})$ to obtain $b_{i,j}$ for all $j \in Adj(i) \cap V_3$

  Decrypts $\bar{s}_{i,j}^{SK}$ with key $f(c_j^{PK}, c_i^{SK})$ to obtain $s_{i,j}^{SK}$ for all $j \in Adj(i) \cap (V_2 \backslash V_3)$

  Sends $\{b_{i,j}\}_{j \in Adj(i) \cap V_3}, \{s_{i,j}^{SK}\}_{j \in Adj(i) \cap (V_2 \backslash V_3)}$ to the server

  **Server:**

  Collects the messages from clients

  Reconstructs $b_i$ from $\{b_{i,j}\}_{j \in Adj(i) \cap V_3}$ for all $i \in V_3$

  Reconstructs $s_i^{SK}$ from $\{s_{i,j}^{SK}\}_{j \in Adj(i) \cap (V_2 \backslash V_3)}$ for all $i \in V_2 \backslash V_3$

  Computes $s_{i,j} = f(s_j^{PK}, s_i^{SK})$ for all $j \in Adj(i) \cap V_3$

  Computes the aggregated sum of local models

$$\sum_{i \in V_3} \theta_i = \sum_{i \in V_3} \tilde{\theta}_i - \sum_{i \in V_3} \textbf{PRG}(b_i) - \sum_{i \in V_2 \backslash V_3, j \in Adj(i) \cap V_3; i > j} \textbf{PRG}(s_{i,j})$$
$$+ \sum_{i \in V_2 \backslash V_3, j \in Adj(i) \cap V_3; i < j} \textbf{PRG}(s_{i,j})$$

---

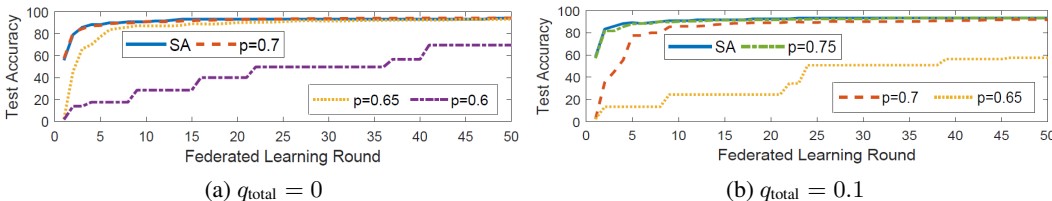

(a) $q_{\text{total}} = 0$                              (b) $q_{\text{total}} = 0.1$

Figure B.1: Test accuracies of SA versus proposed CCESA$(n, p)$ with various connection probability $p$, for federated learning using the AT&T face dataset. Here, we set $n = 40$ and $t = 21$. The suggested CCESA achieves the ideal test accuracy by using only 70% of the communication/computational resources used in the conventional SA.

| Schemes \ Number of training data ($n_{\text{train}}$) | 5000 | 10000 | 15000 | 50000 |
|---|---|---|---|---|
| **Federated Averaging** (McMahan et al., 2017) | 70.41% | 65.82% | 65.89% | 60.62% |
| **Secure Aggregation (SA)** (Bonawitz et al., 2017) | 49.78% | 49.97% | 49.91% | 49.10% |
| **CCESA** (Suggested) | 49.48% | 50.07% | 49.16% | 50.00% |

Table B.1: Precision of the membership inference attack on local models trained on CIFAR-10. The scheme with a higher attack precision is more vulnerable to the inference attack. For the proposed CCESA, the attacker is no better than the random guess with precision = 50%, showing the privacy-preserving ability of CCESA.

# B    ADDITIONAL EXPERIMENTAL RESULTS

## B.1    RELIABILITY

In Fig. 4 of the main paper, we provided the experimental results on the reliaiblity of CCESA on CIFAR-10 dataset. Similarly, Fig. B.1 shows the reliability of CCESA in AT&T Face dataset, where the model is trained over $n = 40$ clients. We plotted the test accuracies of SA and the suggested CCESA$(n, p)$ for various $p$. In both settings of $q_{\text{total}}$, selecting $p = 0.7$ is sufficient to achieve the test accuracy performance of SA when the system is trained for 50 rounds. Thus, the required communication/computational resources for guaranteeing the reliability, which is proportional to $np$, can be reduced to 70% of the conventional wisdom in federated learning.

## B.2    PRIVACY

In Section 5.3 and Fig. 2 of the main paper, we provided the experimental results on the AT&T Face dataset, under the model inversion attack. In Fig. B.2, we provide additional experimental results on the same dataset for different participants. Similar to the result in Fig. 2, the model inversion attack successfully reveals the individuals identity in federated averaging (McMahan et al., 2017), while the privacy attack is not effective in both SA and the suggested CCESA.

In the main manuscript, we have also considered another type of privacy threat called membership inference attack (Shokri et al., 2017), where the attacker observes masked local model $\tilde{\theta}_i$ sent from client $i$ to the server, and guesses whether a particular data is the member of the training set. We measured three types of performance metrics of the attacker: *accuracy* (the fraction of the records correctly estimated the membership), *precision* (the fraction of the responses inferred as members of the training dataset that are indeed members) and *recall* (the fraction of the training data that the attacker can correctly infer as members). Table 3 in the main manuscript summarizes the attack accuracy result, while Table B.1 shows the attack precision for CIFAR-10 dataset. We also observed that recall is close to 1 for all schemes. Similar to the results on the attack accuracy, Table B.1 shows that the attack precision of federated averaging reaches near 70%, while that of SA and CCESA remain around the baseline performance of random guess. This shows that both SA and CCESA do not reveal any clue on the training set.

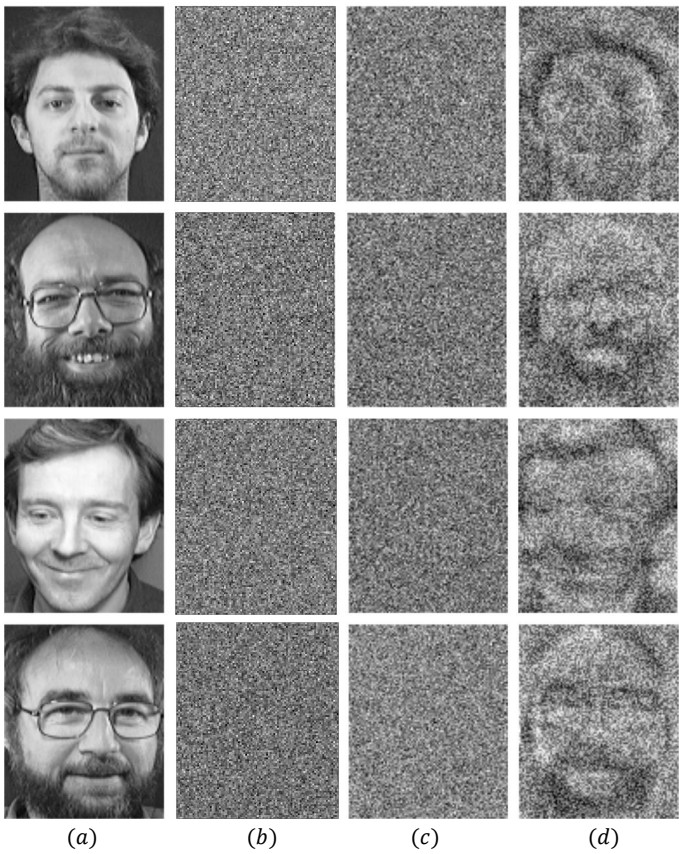

$(a)$ $\qquad$ $(b)$ $\qquad$ $(c)$ $\qquad$ $(d)$

Figure B.2: The result of model inversion attack to three schemes, (b) the suggested scheme (CCESA), (c) SA (Bonawitz et al., 2017) and (d) federated averaging (McMahan et al., 2017), for AT&T Face dataset. The original training images at (a) can be successfully reconstructed by the attack only in federated averaging setup, i.e., both SA and CCESA achieve the same level of privacy.

# C PROOFS

## C.1 PROOF OF THEOREM 1

*Proof.* Note that the sum of masked local models obtained by the server is expressed as

$$\sum_{i \in V_3} \tilde{\theta}_i = \sum_{i \in V_3} \theta_i + \sum_{i \in V_3} \mathbf{PRG}(b_i) + \mathbf{z}$$

where

$$\mathbf{z} = \sum_{i \in V_3} \sum_{j \in V_2 \cap Adj(i); i<j} \mathbf{PRG}(s_{i,j}) - \sum_{i \in V_3} \sum_{j \in V_2 \cap Adj(i); i>j} \mathbf{PRG}(s_{i,j}).$$

Here, $\mathbf{z}$ can be rewritten as

$$\mathbf{z} = \sum_{i \in V_3} \sum_{j \in V_2 \cap Adj(i); i<j} \mathbf{PRG}(s_{i,j}) - \sum_{i \in V_3} \sum_{j \in V_2 \cap Adj(i); i>j} \mathbf{PRG}(s_{i,j})$$

$$= \sum_{i \in V_3} \sum_{j \in V_2 \cap Adj(i); i<j} \mathbf{PRG}(s_{i,j}) - \sum_{j \in V_2} \sum_{i \in V_3 \cap Adj(j); i>j} \mathbf{PRG}(s_{i,j})$$

$$\overset{(a)}{=} \sum_{i \in V_3} \sum_{j \in V_2 \cap Adj(i); i<j} \mathbf{PRG}(s_{i,j}) - \sum_{i \in V_2} \sum_{j \in V_3 \cap Adj(i); i<j} \mathbf{PRG}(s_{i,j})$$

$$= \sum_{i \in V_3} \sum_{j \in V_3 \cap Adj(i); i<j} \mathbf{PRG}(s_{i,j}) + \sum_{i \in V_3} \sum_{j \in (V_2 \setminus V_3) \cap Adj(i); i<j} \mathbf{PRG}(s_{i,j})$$

$$\quad - \sum_{i \in V_3} \sum_{j \in V_3 \cap Adj(i); i<j} \mathbf{PRG}(s_{i,j}) - \sum_{i \in V_2 \setminus V_3} \sum_{j \in V_3 \cap Adj(i); i<j} \mathbf{PRG}(s_{i,j})$$

$$= \sum_{i \in V_3} \sum_{j \in (V_2 \setminus V_3) \cap Adj(i); i<j} \mathbf{PRG}(s_{i,j}) - \sum_{j \in V_3} \sum_{i \in (V_2 \setminus V_3) \cap Adj(j); i<j} \mathbf{PRG}(s_{i,j})$$

$$\overset{(b)}{=} \sum_{i \in V_3} \sum_{j \in (V_2 \setminus V_3) \cap Adj(i); i<j} \mathbf{PRG}(s_{i,j}) - \sum_{i \in V_3} \sum_{j \in (V_2 \setminus V_3) \cap Adj(i); i>j} \mathbf{PRG}(s_{i,j})$$

$$= \sum_{i \in V_3} \sum_{j \in (V_2 \setminus V_3) \cap Adj(i); i<j} \mathbf{PRG}(s_{i,j}) - \sum_{i \in V_3} \sum_{j \in (V_2 \setminus V_3) \cap Adj(i); i>j} \mathbf{PRG}(s_{i,j})$$

$$= \sum_{i \in V_3} \sum_{j \in (V_3^+ \setminus V_3) \cap Adj(i); i<j} \mathbf{PRG}(s_{i,j}) - \sum_{i \in V_3} \sum_{j \in (V_3^+ \setminus V_3) \cap Adj(i); i>j} \mathbf{PRG}(s_{i,j}),$$

where $(a)$ and $(b)$ come from $s_{i,j} = s_{j,i}$. In order to obtain the sum of unmasked local models $\sum_{i \in V_3} \theta_i$ from the sum of masked local models $\sum_{i \in V_3} \tilde{\theta}_i$, the server should cancel out all the random terms in $\sum_{i \in V_3} \mathbf{PRG}(b_i) + \mathbf{z}$. In other words, the server should reconstruct $b_i$ for all $i \in V_3$ and $s_j^{SK}$ for all $j \in V_3^+ \setminus V_3$. Since the server can obtain $|(Adj(i) \cup \{i\}) \cap V_4|$ secret shares of client $i$ in **Step 3**, $|(Adj(i) \cup \{i\}) \cap V_4| \geq t_i$ for all $i \in V_3^+$ is a sufficient condition for reliability.

Now we prove the converse part by contrapositive. Suppose there exists $i \in V_3^+$ such that $|(Adj(i) \cup \{i\}) \cap V_4| < t_i$. In this case, note that the server cannot reconstruct both $s_i^{SK}$ and $b_i$ from the shares. If $i \in V_3$, the server cannot subtract $\mathbf{PRG}(b_i)$ from $\sum_{i \in V_3} \tilde{\theta}_i$. As a result, the server cannot obtain $\sum_{i \in V_3} \theta_i$. If $i \in V_3^+ \setminus V_3$, the server cannot subtract $\mathbf{PRG}(s_{i,j})$ for all $j \in V_3$ since the server does not have any knowledge of neither $s_i^{SK}$ nor $s_j^{SK}$. Therefore, the server cannot compute $\sum_{i \in V_3} \theta_i$, which completes the proof.

$\square$

## C.2 PROOF OF LEMMA 1

*Proof.* Let $\mathcal{T} \subset V_3$ be an arbitrary set of clients satisfying $\mathcal{T} \notin \{\varnothing, V_3\}$. It is sufficient to prove the following statement: given a connected graph $G_3$, an eavesdropper cannot obtain the partial sum of

local models $\sum_{i \in \mathcal{T}} \theta_i$ from the sum of masked models $\sum_{i \in \mathcal{T}} \tilde{\theta}_i$. More formally, we need to prove

$$H(\sum_{i \in \mathcal{T}} \theta_i | \sum_{i \in \mathcal{T}} \tilde{\theta}_i) = H(\sum_{i \in \mathcal{T}} \theta_i).$$

Note that the sum of masked local models $\sum_{i \in \mathcal{T}} \tilde{\theta}_i$ accessible to the eavesdropper is expressed as

$$\sum_{i \in \mathcal{T}} \tilde{\theta}_i = \sum_{i \in \mathcal{T}} \theta_i + \sum_{i \in \mathcal{T}} \mathbf{PRG}(b_i) + \mathbf{z}, \tag{6}$$

where

$$
\begin{aligned}
\mathbf{z} &= \sum_{i \in \mathcal{T}} \sum_{j \in V_2 \cap Adj(i); i<j} \mathbf{PRG}(s_{i,j}) - \sum_{i \in \mathcal{T}} \sum_{j \in V_2 \cap Adj(i); i>j} \mathbf{PRG}(s_{i,j}) \\
&= \sum_{i \in \mathcal{T}} \sum_{j \in V_2 \cap Adj(i); i<j} \mathbf{PRG}(s_{i,j}) - \sum_{j \in V_2} \sum_{i \in \mathcal{T} \cap Adj(j); i>j} \mathbf{PRG}(s_{i,j}) \\
&\overset{(a)}{=} \sum_{i \in \mathcal{T}} \sum_{j \in V_2 \cap Adj(i); i<j} \mathbf{PRG}(s_{i,j}) - \sum_{i \in V_2} \sum_{j \in \mathcal{T} \cap Adj(i); i<j} \mathbf{PRG}(s_{i,j}) \\
&= \sum_{i \in \mathcal{T}} \sum_{j \in \mathcal{T} \cap Adj(i); i<j} \mathbf{PRG}(s_{i,j}) + \sum_{i \in \mathcal{T}} \sum_{j \in (V_2 \backslash \mathcal{T}) \cap Adj(i); i<j} \mathbf{PRG}(s_{i,j}) \\
&\quad - \sum_{i \in \mathcal{T}} \sum_{j \in \mathcal{T} \cap Adj(i); i<j} \mathbf{PRG}(s_{i,j}) - \sum_{i \in V_2 \backslash \mathcal{T}} \sum_{j \in \mathcal{T} \cap Adj(i); i<j} \mathbf{PRG}(s_{i,j}) \\
&= \sum_{i \in \mathcal{T}} \sum_{j \in (V_2 \backslash \mathcal{T}) \cap Adj(i); i<j} \mathbf{PRG}(s_{i,j}) - \sum_{j \in \mathcal{T}} \sum_{i \in (V_2 \backslash \mathcal{T}) \cap Adj(j); i<j} \mathbf{PRG}(s_{i,j}) \\
&\overset{(b)}{=} \sum_{i \in \mathcal{T}} \sum_{j \in (V_2 \backslash \mathcal{T}) \cap Adj(i); i<j} \mathbf{PRG}(s_{i,j}) - \sum_{i \in \mathcal{T}} \sum_{j \in (V_2 \backslash \mathcal{T}) \cap Adj(i); i>j} \mathbf{PRG}(s_{i,j}) \\
&= \{ \sum_{i \in \mathcal{T}} \sum_{j \in (V_2 \backslash V_3) \cap Adj(i); i<j} \mathbf{PRG}(s_{i,j}) - \sum_{i \in \mathcal{T}} \sum_{j \in (V_2 \backslash V_3) \cap Adj(i); i>j} \mathbf{PRG}(s_{i,j}) \} \\
&\quad + \{ \sum_{i \in \mathcal{T}} \sum_{j \in (V_3 \backslash \mathcal{T}) \cap Adj(i); i<j} \mathbf{PRG}(s_{i,j}) - \sum_{i \in \mathcal{T}} \sum_{j \in (V_3 \backslash \mathcal{T}) \cap Adj(i); i>j} \mathbf{PRG}(s_{i,j}) \}.
\end{aligned}
$$

Here, $(a)$ and $(b)$ come from $s_{i,j} = s_{j,i}$. If $G_3 = (V_3, E_3)$ is connected, there exists an edge $e = \{p, q\}$ such that $p \in \mathcal{T}$ and $q \in (V_3 \backslash \mathcal{T})$. As a consequence, pseudorandom term $\mathbf{PRG}(s_{p,q})$ is included in $\mathbf{z}$, and its coefficient $c_{p,q}$ is determined as 1 (if $p < q$), or $-1$ (if $p > q$). Note that equation (6) can be rewritten as

$$\sum_{i \in \mathcal{T}} \tilde{\theta}_i = \sum_{i \in \mathcal{T}} \theta_i + c_{p,q} \cdot \mathbf{PRG}(s_{p,q}) + \mathbf{r},$$

where $\mathbf{r}$ is the sum of pseudorandom terms which do not include $\mathbf{PRG}(s_{p,q})$. In order to unmask $\mathbf{PRG}(s_{p,q})$, the eavesdropper need to know at least one of the secret keys of clients $p$ and $q$. However, the eavesdropper cannot obtain any shares of the secret keys since the server do not request the shares of $s_p^{SK}$ and $s_q^{SK}$ in step 3. Therefore, due to the randomness of pseudorandom generator, $H(\sum_{i \in \mathcal{T}} \theta_i | \sum_{i \in \mathcal{T}} \tilde{\theta}_i) = H(\sum_{i \in \mathcal{T}} \theta_i)$ holds, which completes the proof. $\qquad \square$

### C.3 PROOF OF THEOREM 2

*Proof.* We first prove that the system is private if $\boldsymbol{G} \in \mathcal{G}_C \cup \mathcal{G}_{NI}$. When $\boldsymbol{G} \in \mathcal{G}_C$, the statement holds directly from Lemma 1. Thus, below we only prove for the case of $\boldsymbol{G} \in \mathcal{G}_{NI}$. Note that it is sufficient to prove the following statement: given a graph evolution $\boldsymbol{G} = (G_0, G_1, \cdots, G_4) \in \mathcal{G}_{NI}$, an eavesdropper cannot obtain the partial sum of local models $\sum_{i \in \mathcal{T}} \theta_i$ from the sum of masked models $\sum_{i \in \mathcal{T}} \tilde{\theta}_i$ for every $\mathcal{T} \subset V_3$ satisfying $\mathcal{T} \notin \{V_3, \varnothing\}$. More formally, we need to prove

$$H(\sum_{i \in \mathcal{T}} \theta_i | \sum_{i \in \mathcal{T}} \tilde{\theta}_i) = H(\sum_{i \in \mathcal{T}} \theta_i). \tag{7}$$

When $\mathcal{T} = C_l$ for some $l \in [\kappa]$, there exists $i^\star \in C_l^+$ such that node $i$ is not informative, according to the definition of $\mathcal{G}_{NI}$. Thus, the server (as well as eavesdroppers) cannot reconstruct both $b_i$ and $s_i^{SK}$. Note that the sum of masked models is

$$\sum_{i \in \mathcal{T}} \tilde{\theta}_i = \sum_{i \in \mathcal{T}} \theta_i + \sum_{i \in \mathcal{T}} \mathbf{PRG}(b_i) + \sum_{j \in \mathcal{T}} \sum_{i \in V_2 \cup Adj(j)} (-1)^{\mathbb{1}_{j>i}} \mathbf{PRG}(s_{j,i}), \tag{8}$$

where $\mathbb{1}_A$ is the indicator function which is value 1 when the statement $A$ is true, and 0 otherwise. When $i^\star \in C_l = \mathcal{T}$, we cannot unmask $\mathbf{PRG}(b_{i^\star})$ in this equation. When $i^\star \in C_l^+ \backslash C_l$, there exists $j \in C_l$ such that $\{i^\star, j\} \in E$. Note that the eavesdropper needs to know either $s_j^{SK}$ or $s_{i^\star}^{SK}$, in order to compute $\mathbf{PRG}(s_{j,i^\star})$. Since $i^\star$ is not informative, the eavesdropper cannot get $s_{i^\star}^{SK}$. Moreover, since the server has already requested the shares of $b_j$, the eavesdropper cannot access $s_j^{SK}$. Thus, the eavesdropper cannot unmask $\mathbf{PRG}(s_{j,i^\star})$ from (8). All in all, the eavesdropper cannot unmask at least one pseudorandom term in $\sum_{i \in \mathcal{T}} \tilde{\theta}_i$, proving (7).

When $\mathcal{T} \neq C_l \quad \forall l \in [\kappa]$, there exists an edge $e = \{p, q\}$ such that $p \in \mathcal{T}$ and $q \in (V_3 \backslash \mathcal{T})$. Thus, we cannot unmask $\mathbf{PRG}(s_{p,q})$ from $\sum_{i \in \mathcal{T}} \tilde{\theta}_i$. Following the steps in Section C.2, we have (7).

Now, we prove the converse by contrapositive: if $\boldsymbol{G} = (G_0, G_1, \cdots, G_4) \in \mathcal{G}_D \cap \mathcal{G}_{NI}^c$, then the system is not private. In other words, we need to prove the following statement: if $G_3$ is disconnected and there exists a component $C_l$ such that all nodes in $C_l$ are informative, then the system is not private. Let $\mathcal{T} = C_l$. Then, the eavesdropper obtains

$$\sum_{i \in \mathcal{T}} \tilde{\theta}_i = \sum_{i \in \mathcal{T}} \theta_i + \sum_{i \in \mathcal{T}} \mathbf{PRG}(b_i) + \mathbf{z},$$

where

$$\mathbf{z} = \sum_{i \in \mathcal{T}} \sum_{j \in V_2 \cap Adj(i)} (-1)^{\mathbb{1}_{i>j}} \mathbf{PRG}(s_{i,j}) \stackrel{(a)}{=} \sum_{i \in \mathcal{T}} \sum_{j \in \mathcal{T}} (-1)^{\mathbb{1}_{i>j}} \mathbf{PRG}(s_{i,j}) \stackrel{(b)}{=} 0.$$

Note that $(a)$ holds since $\mathcal{T}$ is a component, $(b)$ holds from $s_{i,j} = s_{j,i}$. Moreover, the eavesdropper can reconstruct $b_i$ for all $i \in \mathcal{T}$ in Step 3 of the algorithm. Thus, the eavesdropper can successfully unmask random terms in $\sum_{i \in \mathcal{T}} \tilde{\theta}_i$ and obtain $\sum_{i \in \mathcal{T}} \theta_i$. This completes the proof. □

### C.4 PROOF OF THEOREM 3

*Proof.* Consider Erdős-Rényi assignment graph $G \in G(n, p)$. Let $N_i := |Adj(i)|$ be the degree of node $i$, and $X_i := |Adj(i) \cap V_4|$ be the number of clients (except client $i$) that successfully send the shares of client $i$ to the server in **Step 3**. Then, $N_i$ and $X_i$ follow the binomial distributions

$$N_i \sim B(n-1, p), \quad X_i \sim B(N_i, (1-q)^4) = B(n-1, p(1-q)^4),$$

respectively. By applying Hoeffding's inequality on random variable $X_i$, we obtain

$$P(X_i < (n-1)p(1-q)^4 - \sqrt{(n-1)\log(n-1)}) \leq 1/(n-1)^2.$$

Let $E$ be the event that the system is not reliable, *i.e.*, the sum of local models $\sum_{i \in V_3} \theta_i$ is not reconstructed by the server, and $E_i$ be the event $\{|(Adj(i) \cup \{i\}) \cap V_4| < t\}$, *i.e.*, a secret of client $i$ is not reconstructed by the server. For a given $p > \frac{t + \sqrt{(n-1)\log(n-1)}}{(n-1)(1-q)^4}$, we obtain

$$P(E) \stackrel{(a)}{=} P(\cup_{i \in V_3^+} E_i) \leq P(\cup_{i \in V_3^+} \{X_i < t\}) \leq \sum_{i \in V_3^+} P(X_i < t)$$

$$\leq \sum_{i \in [n]} P(X_i < t) = nP(X_1 < t)$$

$$\leq nP(X_1 < (n-1)p(1-q)^4 - \sqrt{(n-1)\log(n-1)}) \leq \frac{n}{(n-1)^2} \xrightarrow{n \to \infty} 0,$$

where $(a)$ comes from Theorem 1. Therefore, we conclude that CCESA$(n,p)$ is asymptotically almost surely (a.a.s.) reliable if $p > \frac{t+\sqrt{(n-1)\log(n-1)}}{(n-1)(1-q)^4}$. Furthermore, based on the parameter selection rule of $t$ in Section F, we obtain a lower bound on $p$ as

$$p > \frac{t+\sqrt{(n-1)\log(n-1)}}{(n-1)(1-q)^4} \geq \frac{\frac{(n-1)p+\sqrt{(n-1)\log(n-1)}+1}{2}+\sqrt{(n-1)\log(n-1)}-1}{(n-1)(1-q)^4}.$$

Rearranging the above inequality with respect to $p$ yields

$$p > \frac{3\sqrt{(n-1)\log(n-1)}-1}{(n-1)(2(1-q)^4-1)}.$$

$\square$

## C.5 Proof of Theorem 4

*Proof.* Let $X := |V_3|$ be the number of clients sending its masked local model in **Step 2**. Then, $X$ follows Binomial random variable $B(n,(1-q)^3)$. Given assignment graph $G$ of CCESA$(n,p)$, note that the induced subgraph $G_3 = G - V\backslash V_3$ is an Erdős-Rényi graph $G(X,p)$.

First, we prove

$$P(G_3 \text{ is connected}\big||X-n(1-q)^3| \leq \sqrt{n\ln n}) \xrightarrow{n\to\infty} 1, \tag{9}$$

if $p > p^\star = \frac{\ln(\lceil n(1-q)^3-\sqrt{n\ln n}\rceil)}{\lceil n(1-q)^3-\sqrt{n\ln n}\rceil}(1+\epsilon)$. The left hand side of (9) can be rewritten as

$$P(G_3 \text{ is connected}\big||X-n(1-q)^3| \leq \sqrt{n\ln n})$$
$$= \frac{\sum_{l\in[n(1-q)^3-\sqrt{n\ln n},n(1-q)^3+\sqrt{n\ln n}]} P(X=l)P(G(l,p) \text{ is connected})}{\sum_{l\in[n(1-q)^3-\sqrt{n\ln n},n(1-q)^3+\sqrt{n\ln n}]} P(X=l)}.$$

Here, we use a well-known property of Erdős-Rényi graph: $G(l,p)$ is asymptotically almost surely (a.a.s.) connected if $p > \frac{(1+\epsilon)\ln l}{l}$ for some $\epsilon > 0$. Since $\frac{\ln l}{l}$ is a decreasing function, $G(l,p)$ is a.a.s. connected for all $l \in [n(1-q)^3 - \sqrt{n\ln n}, n(1-q)^3 + \sqrt{n\ln n}]$ when $p > \frac{\ln(\lceil n(1-q)^3-\sqrt{n\ln n}\rceil)}{\lceil n(1-q)^3-\sqrt{n\ln n}\rceil}$. Thus, for given $p > p^\star$, we can conclude

$$P(G_3 \text{ is connected}\big||X-n(1-q)^3| \leq \sqrt{n\ln n}) \xrightarrow{n\to\infty} 1.$$

Now, we will prove that CCESA$(n,p)$ is a.a.s. private when $p > p^\star$. The probability that CCESA$(n,p)$ is private is lower bounded by

$$P(CCESA(n,p) \text{ is private}) \overset{(a)}{\geq} P(G_3 \text{ is connected})$$
$$= P(|X-n(1-q)^3| \leq \sqrt{n\ln n})P(G_3 \text{ is connected}\big||X-n(1-q)^3| \leq \sqrt{n\ln n})$$
$$+ P(|X-n(1-q)^3| > \sqrt{n\ln n})P(G_3 \text{ is connected}\big||X-n(1-q)^3| > \sqrt{n\ln n})$$
$$\overset{(b)}{\geq} (1-2/n^2)P(G_3 \text{ is connected}\big||X-n(1-q)^3| \leq \sqrt{n\ln n}) \xrightarrow{n\to\infty} 1,$$

where $(a)$ comes from Lemma 1 and $(b)$ comes from Hoeffding's inequality

$$P(|X-n(1-q)^3| \leq \sqrt{n\ln n}) \geq 1-2/n^2,$$

which completes the proof. $\square$

## C.6 Proof of Theorem 5

*Proof.* Consider an Erdős-Rényi assignment graph $G \in G(n,p)$. Let $N_i := |Adj(i)|$ be the degree of node $i$, and $X_i := |Adj(i) \cap V_4|$ be the number of clients (except client $i$) that successfully send the shares of client $i$ to the server in **Step 3**. Then, $N_i$ and $X_i$ follow the binomial distributions

$$N_i \sim B(n-1,p), \quad X_i \sim B(N_i,(1-q)^4) = B(n-1,p(1-q)^4),$$

respectively. Let $E_i$ be an event $\{|(Adj(i) \cup \{i\}) \cap V_4| < t\}$, *i.e.*, a secret of client $i$ is not recon-structed by the server. We obtain an upper bound on $P(E_i)$ as

$$P(E_i) \le P(X_i < t) = \sum_{i=0}^{t-1} \binom{n-1}{i}(p(1-q)^4)^i(1-p(1-q)^4)^{(n-1-i)}$$

$$\overset{(a)}{=} e^{-(n-1)D(\frac{t-1}{n-1}||p(1-q)^4)}$$

where $(a)$ comes from Chernoff bound on the binomial distribution. Thus, $P_e^{(r)}$ is upper bounded by

$$P_e^{(r)} \overset{(b)}{=} P(\cup_{i \in V_3^+} E_i) \le P(\cup_{i \in V_3^+}\{X_i < t\}) \le \sum_{i \in V_3^+} P(X_i < t)$$

$$\le \sum_{i \in [n]} P(X_i < t) = nP(X_1 < t) = ne^{-(n-1)D(\frac{t-1}{n-1}||p(1-q)^4)},$$

where $(b)$ comes from Theorem 1. □

## C.7 PROOF OF THEOREM 6

*Proof.* Let $P_{dc}(n, p)$ be the probability of an event that Erdős-Rényi graph $G \in G(n, p)$ is discon-nected. Then, $P_{dc}(n, p)$ is upper bounded as follows.

$$P_{dc}(n, p) = P(G(n, p) \text{ is disconnected})$$

$$= P(\cup_{k=1}^{\lfloor n/2 \rfloor}\{\text{there exists a subset of } k \text{ nodes that is disconnected}\})$$

$$\le \sum_{k=1}^{\lfloor n/2 \rfloor} P(\text{there exists a subset of } k \text{ nodes that is disconnected})$$

$$\le \sum_{k=1}^{\lfloor n/2 \rfloor} \binom{n}{k} P(\text{a specific subset of } k \text{ nodes is disconnected})$$

$$= \sum_{k=1}^{\lfloor n/2 \rfloor} \binom{n}{k}(1-p)^{k(n-k)}$$

Therefore, $P_e^{(p)}$ is upper bounded by

$$P_e^{(p)} \overset{(a)}{\le} P(G_3 = G - V\backslash V_3 \text{ is disconnected})$$

$$= \sum_{m=0}^{n} P(G_3 \text{ has } m \text{ vertices})P_{dc}(m, p)$$

$$= \sum_{m=0}^{n} \binom{n}{m}(1-q)^{3m}(1-(1-q)^3)^{(n-m)} \cdot P_{dc}(m, p)$$

$$= \sum_{m=0}^{n} \binom{n}{m}(1-q)^{3m}(1-(1-q)^3)^{(n-m)} \sum_{k=1}^{\lfloor m/2 \rfloor} \binom{m}{k}(1-p)^{k(m-k)},$$

where $(a)$ comes from Lemma 1. □

## D REQUIRED RESOURCES OF CCESA

### D.1 COMMUNICATION COST

Here, we derive the additional communication bandwidth $B_{CCESA}$ used at each client for running CCESA, compared to the bandwidth used for running federated averaging (McMahan et al., 2017).

We consider the worst-case scenario having the maximum additional bandwidth, where no client fails during the operation.

The required communication bandwidth of each client is composed of four parts. First, in **Step 0**, each client $i$ sends two public keys to the server, and receives $2|Adj(i)|$ public keys from other clients. Second, in **Step 1**, each client $i$ sends encrypted $2|Adj(i)|$ shares to other nodes, and receives $2|Adj(i)|$ shares from other nodes through the server. Third, in **Step 2**, each client $i$ sends a masked data $y_i$ of $mR$ bits. Here, $m$ is the dimension of model parameters where each parameter is represented in $R$ bits. Fourth, in **Step 3**, each client $i$ sends $|Adj(i)| + 1$ shares to the server. Therefore, total communication bandwidth of client $i$ can be expressed as

$$(\text{total communication bandwidth}) = 2(|Adj(i)| + 1)a_K + (5|Adj(i)| + 1)a_S + mR,$$

where $a_K$ and $a_S$ are the number of bits required for exchanging public keys and the number of bits in a secret share. Since each client $i$ requires $mR$ bits to send the private vector $\theta_i$ in the federated averaging (McMahan et al., 2017), we have

$$B_{CCESA} = 2(|Adj(i)| + 1)a_K + (5|Adj(i)| + 1)a_S.$$

If we choose the connection probability $p = (1 + \epsilon)p^\star$ for a small $\epsilon > 0$, we have $B_{CCESA} = O(\sqrt{n \log n})$, where $p^\star$ is defined in (5). Note that the additional bandwidth $B_{SA}$ required for SA can be similarly obtained as $B_{SA} = 2na_K + (5n - 4)a_S$ having $B_{SA} = O(n)$. Thus, we have

$$\frac{B_{CCESA}}{B_{SA}} \to 0$$

as $n$ increases, showing the scalability of CCESA. These results are summarized in Table 1 in the main manuscript.

### D.2 Computational cost

We evaluate the computational cost of CCESA. Here we do not count the cost for computing the signatures since it is negligible. First, we derive the computational cost of each client. Given the number of model parameters $m$ and the number of clients $n$, the computational cost of client $i$ is composed of three parts: (1) computing $2|Adj(i)|$ key agreements, which takes $O(|Adj(i)|)$ time, (2) generating shares of $t_i$-out-of-$|Adj(i)|$ secret shares of $s_i^{SK}$ and $b_i$, which takes $O(|Adj(i)|^2)$ time, and (3) generating masked local model $\tilde{\theta}_i$, which requires $O(m|Adj(i)|)$ time. Thus, the total computational cost of each client is obtained as $O(|Adj(i)|^2 + m|Adj(i)|)$. Second, the server's computational cost is composed of two parts: (1) reconstructing $t_i$-out-of-$|Adj(i)|$ secrets from shares for all clients $i \in [n]$, which requires $O(|Adj(i)|^2)$ time, and (2) removing masks from masked sum of local models $\sum_{i=1}^{n} \tilde{\theta}_i$, which requires $O(m|Adj(i)|^2)$ time in the worst case. As a result, the total computational cost of the server is $O(m|Adj(i)|^2)$. If we choose $p = (1 + \epsilon)p^\star$ for small $\epsilon > 0$, the total computational cost per each client is $O(n \log n + m\sqrt{n \log n})$, while the total computation cost of the server is $O(mn \log n)$. The computational cost of SA can be obtained in a similar manner, by setting $Adj(i) = n - 1$; each client requires $O(n^2 + mn)$ time while the server requires $O(mn^2)$ time. These results are summarized in Table 1 in the main manuscript.

## E Reliability and privacy of CCESA

Here, we analyze the asymptotic behavior of probability that a system is reliable/private. In our analysis, we assume that the connection probability is set to $p^\star$ and the parameter $t$ used in the secret sharing is selected based on the rule in Section F. First, we prove that a system is reliable with probability $\geq 1 - O(ne^{-n \log n})$. Using Theorem 5, the probability that a system is reliable can be directly derived as

$$P(\text{A system is reliable}) = 1 - P_e^{(r)}$$

$$\geq 1 - ne^{-(n-1)D_{KL}(\frac{t-1}{n-1}||p^\star(1-q)^4)}.$$

Using the fact that Kullback-Leibler divergence term satisfies

$$D_{KL}\Big(\frac{t-1}{n-1}||p^\star(1-q)^4\Big) = \frac{t-1}{n-1}\log\Big(\frac{\frac{t-1}{n-1}}{p^\star(1-q)^4}\Big) + \Big(1 - \frac{t-1}{n-1}\Big)\log\Big(\frac{1 - \frac{t-1}{n-1}}{1 - p^\star(1-q)^4}\Big)$$

$$= \Theta(\sqrt{\log n/n}),$$

we conclude that CCESA$(n, p^\star)$ is reliable with probablilty $\geq 1 - O(ne^{-\sqrt{n \log n}})$.

Now we prove that a system is private with probability $\geq 1 - O(n^{-C})$ for an arbitrary $C > 0$. Using Theorem 6, the probability that a system is private can be obtained as

$$P(\text{A system is private}) = 1 - P_e^{(p)}$$
$$\geq 1 - \sum_{m=0}^{n} a_m b_m,$$

where $a_m = \binom{n}{m}(1 - q)^{3m}(1 - (1 - q)^3)^{(n-m)}$ and $b_m = \sum_{k=1}^{\lfloor m/2 \rfloor} \binom{m}{k}(1 - p^\star)^{k(m-k)}$. Note that the summation term $\sum_{m=0}^{n} a_m b_m$ can be broken up into two parts: $\sum_{m=0}^{m_{th}} a_m b_m$ and $\sum_{m=m_{th}+1}^{n} a_m b_m$, where $m_{th} = \lfloor n(1 - q)^3/2 \rfloor$. In the rest of the proof, we will prove two lemmas, showing that $\sum_{m=0}^{m_{th}} a_m b_m = O(e^{-n})$ and $\sum_{m=m_{th}+1}^{n} a_m b_m = O(n^{-C})$, respectively.

**Lemma 2.**

$$\sum_{m=0}^{m_{th}} a_m b_m = O(e^{-n})$$

*Proof.* Since $b_m \leq 1$ for all $m$, we have

$$\sum_{m=0}^{m_{th}} a_m b_m \leq \sum_{m=0}^{m_{th}} a_m.$$

Note that $a_m = P(X = m)$ holds for binomial random variable $X = B(n, (1 - q)^3)$. By utilizing Hoeffding's inequality, we have

$$\sum_{m=0}^{m_{th}} a_m = P(X \leq m_{th}) \leq e^{-2(n(1-q)^3 - m_{th})^2} \leq e^{-n(1-q)^6/2}.$$

Therefore, we conclude that $\sum_{m=0}^{m_{th}} a_m b_m = O(e^{-n})$. $\qquad\square$

**Lemma 3.**

$$\sum_{m=m_{th}+1}^{n} a_m b_m = O(n^{-C})$$

*Proof.* Since $a_m \leq 1$ for all $m$, we have

$$\sum_{m=m_{th}+1}^{n} a_m b_m \leq \sum_{m=m_{th}+1}^{n} b_m.$$

Let $C > 0$ be given. Then, the upper bound on $b_m$ can be obtained as

$$b_m = \sum_{k=1}^{\lfloor m/2 \rfloor} \binom{m}{k}(1 - p^\star)^{k(m-k)} \leq \sum_{k=1}^{\lfloor m/2 \rfloor} \binom{m}{k} e^{-k(m-k)p^\star}$$
$$= \sum_{k=1}^{\lfloor m/2 \rfloor} \binom{m}{k} m^{-\lambda k(m-k)/m} = c_m + d_m$$

where $\lambda = p^\star n/\log n$, $c_m = \binom{m}{k}\sum_{k=1}^{k^\star} m^{-\lambda k(m-k)/m}$, $d_m = \binom{m}{k}\sum_{k=k^\star+1}^{\lfloor m/2 \rfloor} m^{-\lambda k(m-k)/m}$, and $k^\star = \lfloor m(1 - \frac{C+2}{\lambda}) \rfloor$ for some $C > 0$. The first part of summation is upper bounded by

$$c_m = \sum_{k=1}^{k^\star} \binom{m}{k} m^{-\lambda k(m-k)/m} \leq \sum_{k=1}^{k^\star} m^{-k\lfloor \lambda(m-k)/m-1 \rfloor} \leq \sum_{k=1}^{k^\star} m^{-k\lfloor \lambda(m-k^\star)/m-1 \rfloor}$$
$$\leq \frac{m^{-\lfloor \lambda(m-k^\star)/m-1 \rfloor}}{1 - m^{-\lfloor \lambda(m-k^\star)/m-1 \rfloor}} = \frac{m^{-(C+1)}}{1 - m^{-(C+1)}}.$$

The second part of summation, we will use the bound $\binom{n}{k} \leq (\frac{en}{k})^k$. Using this bound, $d_m$ is upper bounded by

$$d_m = \sum_{k=k^\star+1}^{\lfloor m/2 \rfloor} \binom{m}{k} m^{-\lambda k(m-k)/m} \leq \sum_{k=k^\star+1}^{\lfloor m/2 \rfloor} (\frac{em^{1-\lambda(m-k)/m}}{k})^k \leq \sum_{k=k^\star+1}^{\lfloor m/2 \rfloor} (\frac{em^{1-\lambda(m-k)/m}}{k^\star+1})^k$$

$$\leq \sum_{k=k^\star+1}^{\lfloor m/2 \rfloor} (\frac{em^{-\lambda(m-k)/m}}{1-\lambda^{-1}(C+2)})^k \leq \sum_{k=k^\star+1}^{\lfloor m/2 \rfloor} (\frac{em^{-\lambda/2}}{1-\lambda^{-1}(C+2)})^k.$$

For sufficiently large $\lambda$, we have $em^{-\lambda/2}/(1-\lambda^{-1}(C+2)) < \delta$ for some $\delta < 1$. Therefore, $d_m$ is upper bounded by

$$d_m \leq \sum_{k=k^\star+1}^{\infty} \delta^k = \frac{\delta^{k^\star}}{1-\delta} = O(\delta^{mC'})$$

where $C' = (1-\lambda^{-1}(C+2)) > 0$. Combining upper bounds on $c_m$ and $d_m$, we obtain $b_m = O(m^{-(C+1)})$. Since $b_m$ is a decreasing function of $m$,

$$\sum_{m=m_{th}+1}^{n} b_m \leq \sum_{m=m_{th}+1}^{n} b_{m_{th}+1} = (n-m_{th})b_{m_{th}+1} \overset{(a)}{=} O(n^{-C})$$

holds where $(a)$ comes from $m_{th} = \lfloor n(1-q)^3/2 \rfloor$.

$\square$

Combining the above two lemmas, we conclude that CCESA$(n, p^\star)$ is private with probability $\geq 1 - O(n^{-C})$ for arbitrary $C > 0$. These results on the reliability and the privacy are summarized in Table 1 of the main manuscript.

## F    DESIGNING THE PARAMETER $t$ FOR THE SECRET SHARING

Here we provide a rule for selecting parameter $t$ used in the secret sharing. In general, setting $t$ to a smaller number is better for tolerating dropout scenarios. However, when $t$ is excessively small, the system is vulnerable to the *unmasking attack* of adversarial server; the server may request shares of $b_i$ and $s_i^{SK}$ to disjoint sets of remaining clients simultaneously, which reveals the local model $\theta_i$ to the server. The following proposition provides a rule of designing parameter $t$ to avoid such unmasking attack.

**Proposition 1** (Lower bound on $t$). *For CCESA$(n, p)$, let $t > \frac{(n-1)p+\sqrt{(n-1)\log(n-1)}+1}{2}$ be given. Then, the system is asymptotically almost surely secure against the unmasking attack.*

*Proof.* Let $E$ be the event that at least one of local models are revealed to the server, and $E_i$ be the event that $i^{\text{th}}$ local model $\theta_i$ is revealed to the server. Note that $\theta_i$ is revealed to the server if $t$ clients send the shares of $b_i$ and other $t$ clients send the shares of $s_i^{SK}$ in **Step 3**. Therefore,

$$P(E_i) \leq P(|(Adj(i) \cup \{i\}) \cap V_4| \geq 2t)$$
$$\leq P(|(Adj(i) \cup \{i\})| \geq 2t) = P(|Adj(i)| \geq 2t-1)$$
$$\leq P(|Adj(i)| > (n-1)p + \sqrt{(n-1)\log(n-1)}) \overset{(a)}{\leq} \frac{1}{(n-1)^2},$$

where (a) comes from Hoeffding's inequality of binomial random variable. As a result, we obtain

$$P(E) = P(\cup_{i \in [n]} E_i) \leq \sum_{i \in [n]} P(E_i) = nP(E_1) = \frac{n}{(n-1)^2} \xrightarrow{n \to \infty} 0,$$

which completes the proof.

$\square$

As stated above, setting $t$ to a smaller number is better to tolerate the dropout of multiple clients. Thus, as in the following remark, we set $t$ to be the minimum value avoiding the unmasking attack.

**Remark 4** (Design rule for $t$). *Throughout the paper, we set $t = \lceil \frac{(n-1)p + \sqrt{(n-1)\log(n-1)} + 1}{2} \rceil$ for CCESA($n, p$), in order to secure a system against the unmasking attack and provide the maximum tolerance against dropout scenarios.*

## G    DETAILED EXPERIMENTAL SETUP

### G.1    AT & T FACE DATASET

AT&T Face dataset contains images of $40$ members. We allocated the data to $n = 40$ clients participating in the federated learning, where each client contains the images of a specific member. This experimental setup is suitable for the practical federated learning scenario where each client has its own image and the central server aggregates the local models for face recognition. Following the previous work (Fredrikson et al., 2015) on the model inversion, we used softmax regression for the classification. Both the number of local training epochs and the number of global aggregation rounds are set to $E_{local} = E_{global} = 50$, and we used the SGD optimizer with learning rate $\gamma = 0.05$.

### G.2    CIFAR-10 DATASET

#### G.2.1    RELIABILITY EXPERIMENT IN FIG. 4

We ran experiments under the federated learning setup where 50000 training images are allocated to $n = 1000$ clients. Here, we considered two scenarios for data allocation: one is partitioning the data in the i.i.d. manner (i.e., each client randomly obtains 50 images), while the other is non-i.i.d. allocation scenario. For the non-i.i.d. scenario, we followed the procedure of (McMahan et al., 2017). Specifically, the data is first sorted by its category, and then the sorted data is divided into 2000 shards. Each client randomly chooses 2 shards for its local training data. Since each client has access to at most 2 classes, the test accuracy performance is degraded compared with the i.i.d. setup. For training the classifier, we used VGG-11 network and the SGD optimizer with learning rate $\gamma = 0.1$ and momentum $\beta = 0.5$. The local training epoch is set to $E_{local} = 3$.

#### G.2.2    PRIVACY EXPERIMENTS IN TABLE 3 AND TABLE B.1

We conducted experiments under the federated learning setup where $n_{\text{train}}$ training images are assigned to $n = 10$ clients. We considered i.i.d. data allocation setup where each client randomly obtains $n_{\text{train}}/10$ training images. The network architecture, the optimizer, and the number of local training epochs are set to the options used in Sec. G.2.1.

### G.3    CONNECTION PROBABILITY SETUP IN FIG. 3

In Fig. 3, we select different connection probabilities $p = p^\star(n, q_{\text{total}})$ for various $n$ and $q_{\text{total}}$, where $p^\star$ is defined in (5). The detailed values of connection probability $p$ are provided in Table G.2.

| $q_{\text{total}} \backslash n$ | 100 | 200 | 300 | 400 | 500 | 600 | 700 | 800 | 900 | 1000 |
|---|---|---|---|---|---|---|---|---|---|---|
| 0 | 0.636 | 0.484 | 0.411 | 0.365 | 0.333 | 0.308 | 0.289 | 0.273 | 0.260 | 0.248 |
| 0.01 | 0.649 | 0.494 | 0.419 | 0.373 | 0.340 | 0.315 | 0.295 | 0.280 | 0.265 | 0.254 |
| 0.05 | 0.707 | 0.538 | 0.457 | 0.406 | 0.370 | 0.344 | 0.321 | 0.304 | 0.289 | 0.276 |
| 0.1 | 0.795 | 0.605 | 0.513 | 0.456 | 0.416 | 0.385 | 0.361 | 0.341 | 0.325 | 0.311 |

Table G.2: Connection probability $p = p^\star$ in Fig. 3

### G.4    RUNNING TIME EXPERIMENT IN TABLE 2

We implemented CCESA algorithm in python. For symmetric authenticated encryption, we use AES-GCM with 128-bit keys in `Crypto.Cipher` package. For the pseudorandom generator, we use `randint` function (input: random seed, output: random integer in the field of size $2^{16}$) in `numpy.random` package. For key agreement, we use Elliptic-Curve Diffie-Hellman over the NIST SP800-56 curve composed with a SHA-256 hash function. For secret sharing, we use standard $t$-out-of-$n$ secret sharing (Shamir, 1979).

