# OpenReview forum: "Communication-Computation Efficient Secure Aggregation for Federated Learning"
_ICLR.cc/2021/Conference — Reject_

### Official Review · AnonReviewer2 · 2020-10-20
**An Ok paper. Unrealistic assumptions.**

**Rating:** 4
**Confidence:** 3

**Review:**

This paper proposes a secure (or private) algorithm that is communication and computationally efficient for the purpose of federated learning. The main idea is to allow each client to share its public keys and shares them secretly with a subset of its clients. This is a very natural idea. The main theoretical contribution is potentially useful. However, it is based on a very unrealistic model of a graph -- namely the Erdos-Renyi (ER) graph. It is widely acknowledged that in the real world, ER graphs are not realistic and indeed, are too simple to model real world networks. Consequently, the analysis leading to Theorems 3 and 4 are overly simplified and the insights on p^* may not carry over to real-world settings.

The analytical results illustrated in Fig 3 are also misleading. The authors mention that the privacy error probability is approx 10^{-40}, but this number depends on what p you choose (that is above p^*). If p is only slightly above p^*, this small error probability is unlikely to materialize. Thus, the discussion here is weak.

There is no impossibility result or any discussion on whether the results in Theorems 3 to 6 are tight in any way. If they were, the illustrations would be more informative.

In the real-world experiments on CIFAR-10, it is mentioned that p = p^* = 0.795 is the provably minimum connection probably. Isn't this number way too large? Essentially your whole graph is connected and the structure, if any, can't be reasonably exploited. This leads me to think that there is a great deal of looseness in the analyses.

---

> ### Author Response · Authors · 2020-11-17
> **Response to Reviewer 2**
>
> **1.(a) Erdos-Renyi (ER) graph is unrealistic model for real world networks**
> > Overall, we disagree that the use of the ER graph is not realistic. Let us clarify the usage of the ER graph in our paper: we are not modeling real-world networks, but designing the key sharing method using ER graphs. It does not limit our algorithm in real-world networks.
> Moreover, regarding the feasibility of our scheme, we have plenty of experiments that show the feasibility of CCESA in practical federated learning setup using real datasets.
>
> **1.(b) Fig.3 is misleading: privacy error probability may not be approximated to** $10^{-40}$  **if** $p$ **is slightly above** $p^*$
> > In the revised manuscript, we modified Fig.3 to avoid misleading the readers. To be specific, let $f(p)$ be the upper bound on the privacy error probability in Theorem 6. In Fig.3 of the revised manuscript, we plotted $f(p^*)$, which is an upper bound on the privacy error probability for all $p>p^*$ since $f$ is a decreasing function. From this revised figure, one can confirm that privacy error probability is upper bounded by $10^{-40}$ for arbitrary $p>p^*$.
>
>
> **2. No discussions on tightness of bounds in Theorems**
> > Although we do not know the tightness of the upper bounds in Theorems 5 and 6, Fig. 3 shows that the upper bound approaches zero in the practical regime. Thus, we feel that our bounds in Theorems 5 and 6 are quite meaningful.
>
>
> **3. In CIFAR-10 experiments, the minimum connection probability** $p^*=0.795$ **is too large**
> > Note that $p^*$ decreases as a function of $n$, the number of clients. When $n=1000$, we have $p^*=0.3106$, and thus the communication and computation cost can be reduced up to 70% as in Fig.4 of the revised manuscript.

---

### Official Review · AnonReviewer3 · 2020-10-27
**I would recommend weak accept. The paper studies an important topic in private federated learning, i.e., improving the communication/computational efficiency. The main concern that limits my score is its novelty/contribution (See reviews).**

**Rating:** 6
**Confidence:** 3

**Review:**

Summary:

The paper proposes a secure aggregation framework for federated learning that is communication-computation efficient. Specifically, instead of sharing its public keys and secret shares to all the other clients as done in the existing scheme, each client only shares to a subset of selected clients, which reduces the communication and computational costs. The sufficient condition on the graph topology of the selected pairs (assignment graph) is identified under which the private and reliable learning is guaranteed. Experiments on real-world datasets validate the theory.


Strength:

The paper provides the rigorous theoretical guarantee for the algorithm, including the followings,
1. Sufficient conditions on the assignment graph are identified under which the federated learning is reliable and private;
2. For Erdos-Renyi random graph where two nodes are connected with probability $p$, the lower bound of $p$ is given such that the algorithm is asymptotically almost surely reliable and private. The upper bound of error probability is also given for a finite number of nodes such that the algorithm is reliable and private.


Weakness/Comments:

1. My biggest concern is the novelty of the paper, which seems to be not significant. Specifically, the proposed algorithm's main contribution is generalizing the existing secure aggregation framework (Bonawitz et al., 2017) from the complete assignment graph to an arbitrary graph. The only modification is on the assignment graph, while the framework itself is still the same as (Bonawitz et al., 2017). Moreover, the idea of limiting communications over a li
distributed learning over the Erdos-Renyi random graph has been studied and analyzed extensively in the literature.

2. In experiments (Fig 4), when p >0.795, the proposed algorithm can achieve the same test accuracy as SA. It seems communication/computational efficiency can be attained "for free. " However, intuitively, if the fewer nodes are aggregated each round in update, the convergence rate ought to decrease. Is the same test accuracy because of the generalization property? How does the training accuracy of two algorithms compared to each other?

3. Related work: secure multiparty computation (MPC) has been studied extensively in distributed learning, with or without a central server. Federated learning essentially is a special type of distributed learning. I suggest authors can include more related work about MPC in distributed learning. For example,
(1) K. Tjell and R. Wisniewski, "Privacy Preservation in Distributed Optimization via Dual Decomposition and ADMM," 2019 IEEE 58th Conference on Decision and Control (CDC), Nice, France, 2019
(2) C. Zhang, M. Ahmad and Y. Wang, "ADMM Based Privacy-Preserving Decentralized Optimization," in IEEE Transactions on Information Forensics and Security, vol. 14, no. 3, pp. 565-580, March 2019
(3) Shen, S., Zhu, T., Wu, D., Wang, W. and Zhou, W., "From distributed machine learning to federated learning: In the view of data privacy and security. Concurrency and Computation: Practice and Experience", 2020.

---

> ### Author Response · Authors · 2020-11-17
> **Response to Reviewer 3**
>
> **1.(a) Novelty is not significant compared with (Bonawitz et al., 2017)**
> > CCESA is based on secure aggregation framework, and necessarily shares common ground with (Bonawitz et al., 2017). However, we have successfully overcome significant theoretical challenges to show that sparse assignment graphs (with appropriate connection probability) guarantee the reliability and privacy conditions.
> Moreover, unlike (Bonawitz et al., 2017), we also applied our algorithm to the federated learning framework, and provided experimental results on the reliability and the privacy of our scheme in real-world scenarios.
>
>
> **1.(b) Compare with related works using Erdos-Renyi graph**
> > In Sec.1 of the revised manuscript, we cited some recent works using Erdos-Renyi (ER) random graph in distributed learning setup. Although these papers and ours share the concept of using the ER graph, our work is the first effort to use this graph for key sharing to enable secure federated learning.
>
>
> **2. Why CCESA with $p >0.795$ in Fig.4 achieves the same test accuracy as SA?**
> > For a fixed dropout probability q, the set of contacted nodes $V_3$ for aggregation at each round is identical in both SA and CCESA with $p > 0.795$. Therefore, we get the identical global model $\sum_{i \in V_3} \theta_i$ (and thus the same training/test accuracies of the model) in both schemes.
>
>
> **3. Related works on secure multiparty computation (MPC) in distributed/decentralized learning**
> > Thank you for the suggestions. We have added citations for these works in Sec. 1 of the revised manuscript.

---

### Official Review · AnonReviewer4 · 2020-10-29
**Official Blind Review #4**

**Rating:** 3
**Confidence:** 5

**Review:**

This paper considers the problem of secure aggregation for federated learning, where the goal is to design a protocol that allows the server to aggregate models from clients without learning anything about any individual model. The paper builds up on the secure aggregation scheme of (Bonawitz et al., 2017), and proposes a scheme that requires smaller communication and computation costs. The main idea is to use a sparse random graph as a communication graph as opposed to the complete graph used by (Bonawitz et al., 2017).

Strong points:

1. Reducing communication and computation costs in secure aggregation is indeed an important practical challenge. The idea of using a sparse communication graph is interesting.

2. The paper gives experimental results on CIFAR-10 and compares with (Bonawitz et al., 2017). On the other hand, (Bonawitz et al., 2017) only present results on synthetic vectors and do not consider any machine learning task in their experiments.

Major concerns:

1. My main concern is that the paper lacks the mathematical rigor required in a theoretical security/privacy paper.

(a) In Definition 2, the eavesdropper model (or threat model) has not been properly defined. In particular, what messages of the protocol the eavesdropper can observe is not explicitly defined. Further, it is not mathematically rigorous to simply say that “an eavesdropper cannot obtain any information on the partial sum”. This needs to be quantified, e.g., by using an information-theoretic or computational expression.

(b) The proof of Theorem 2 in supplementary material simply shows that the eavesdropper cannot compute the sum of local models $\sum_{i\in\mathcal{T}} \theta_i$ from the sum of the masked models $\sum_{i\in\mathcal{T}} \tilde{\theta}_i$. How does this guarantee that the eavesdropper cannot obtain ‘any information’ about the sum of local models? The proof is not rigorous (partially due to the ill-defined privacy requirement).

(c) Sufficient details are not provided in experiments. Specifically, what encryption scheme is used, what PRG is used?

2. From the proof of Theorem 2, it is assumed that the eavesdropper has access to masked local models of a subset $\mathcal{T} \subset V_3$ of nodes. This is a much weaker adversary model than (Bonawitz et al., 2017). It does not seem fair to compare costs of protocols that give security guarantees for different threat models. At the very least, it should be explicitly mentioned upfront that  the proposed protocol is secure against a weaker threat model than (Bonawitz et al., 2017).

3. Similarly, the attack considered in experiments (Sec. 5.3) is fairly simple. Secure aggregation schemes in (Bonawitz et al., 2017) and (So et al., 2020) provide security against much stronger attacks — privacy is guaranteed even if a subset of devices collude with each other, or the server colludes with a subset of devices.

4. The following recent paper uses a very similar idea to reduce communication and computation costs of secure aggregation. (This paper gives precise privacy definitions and rigorous mathematical proofs for security.)

James Bell, K. A. Bonawitz, Adrià Gascón, Tancrède Lepoint, Mariana Raykova, “Secure Single-Server Aggregation with (Poly)Logarithmic Overhead”, Jun 2020. (https://eprint.iacr.org/2020/704)

Even if one considers CCESA as a parallel and independent work, it will be helpful to acknowledge the above paper.

Overall, the paper proposes an interesting idea of using sparse communication graphs for secure aggregation to reduce communication and computation costs. However, the paper seems to lack the mathematical rigor and details.

----------- Post-Rebuttal Comments -----------------
Thanks to the authors for the response and for updating the draft. Some of my queries were clarified. However, I still think the paper lacks the mathematical rigor required for a theoretical security/privacy paper. For instance, in the updated proof of Theorem 2, the authors consider the output of a pseudo-random generator (PRG) as uniformly random and claim information-theoretic (perfect) security. However, PRG output is not uniformly random, and one needs to consider computational security, which is standard in cryptography/security literature. Moreover, the comparison with (Bonawitz et al., 2017) and (Bell et al., 2020) seems a bit unfair since the threat model in the paper is weaker. As an example, for the same threat model in the paper, it is not clear if (Bell et al., 2020) would need the strong assumption on dropouts. For these reasons, I retain my original score.

---

> ### Author Response · Authors · 2020-11-17
> **Response to Reviewer 4**
>
> **1.(a)-1 Need clear definition on the threat model**
> > As added in page 5 of the revised manuscript, the threat model can be clarified as follows: an eavesdropper can access the information transmitted between any client and the server from Step 0 to Step 3 of CCESA algorithm, namely, public keys, secret shares, masked local models and the indices of survived clients $V_3$.
>
> **1.(a)-2 Need rigorous privacy argument**
> > The private system can be defined using the information-theoretic terminology: Let $E$ be the information accessible to the eavesdropper, and $H$ be the entropy function. Then, the system is called “private” if $H(\sum_{i \in T} \theta_i) = H(\sum_{i \in T} \theta_i \lvert E)$ holds for any $T\subset V_3$ satisfying $T \notin$ {$V_3,\varnothing$}. We have clarified our threat model and privacy argument in Definition 2 of the revised manuscript accordingly.
>
>
> **1.(b) Need rigorous proof of Theorem 2: why the eavesdropper cannot obtain any information about the sum of local models?**
> > The eavesdropper cannot obtain any information about the sum of local models due to the randomness of the pseudorandom generator. To be specific, note that the sum of masked local models obtained by an eavesdropper contains at least one completely random term **PRG**$(s_{p,q})$ that the eavesdropper cannot unmask. As a consequence, no information on the sum of local models is revealed to the eavesdropper. We have included a more detailed proof in Section C.2 of the supplementary materials of the revised manuscript.
>
>
> **1.(c) Detail experimental setup (encryption, PRG)**
> > For symmetric authenticated encryption, we used AES-GCM with 128-bit keys in `Crypto.Cipher` package in python. For the pseudorandom generator, we used the `randint` function (input: random seed, output: random integer in the field of size $2^{16}$) in `numpy.random` package. We provided detailed experimental setups in Sec.G.4 of the supplementary materials of the revised manuscript.
>
>
> **2. Threat model is weaker than that of (Bonawitz et al. 2017.)**
> > We agree that our threat model is weaker than (Bonawitz et al, 2017). To be specific, our threat model is equivalent to the “server-only” honest-but-curious adversary without colluding clients, while the threat model in (Bonawitz et al, 2017) is “server-client collusion” honest-but-curious adversary. Note that we target protection against eavesdroppers, and our threat model simply reflects that. We explicitly mentioned this difference in page 5 of the revised manuscript.
>
>
> **3. Simple attack used in privacy experiments: (Bonawitz et al., 2017) and (So et al., 2020) considered stronger attacks**
> > Although the authors of (Bonawitz et al., 2017) and (So et al., 2020) considered diverse attack scenarios, they did not demonstrate the security of their schemes through experiments on privacy attacks. Note that the experimental results in Sec.5.3 of our paper are quite meaningful in that the performance has been validated under popular privacy threats (membership inference attack and model inversion attack) in federated learning environments.
>
>
> **4. Comments on a very recent related work (Bell et al. 2020.)**
> > Thank you for informing the presence of this work; we have cited it in Sec. 1 of the revised manuscript. We remark that our work is unique in that performance has been validated in federated learning environments. Also, our work does not assume anything on the number of clients dropped out of the protocol, while the work of Bell et al. is based on a strong assumption on the number of dropouts.

---

### Official Review · AnonReviewer1 · 2020-10-31
**A nice idea with insufficient analysis**

**Rating:** 4
**Confidence:** 4

**Review:**

This paper proposes an efficiency improvement on the "secure aggregation" (called SA in this paper) protocol of Bonawitz et al.

For context: the SA protocol allows a group of clients holding secret values to compute the sum of their values with the help of a reliable server. The computation is "private" in the following sense: any adversary that controls the server and at most $t$ out of the $n$ clients learns nothing except the sum of the honest parties' inputs.

More formally: SA is a multiparty computation protocol for the "summation" ideal functionality, that is secure against a *semi-honest* honest adversary. It has the desirable feature of being resilient to failures on the part of many clients (as long as the server remains online).

The big downsides are that (a) the protocol does not handle malicious behavior on the part of the server and clients and (b) the communication scales poorly with the number $n$ of clients (each client's communication is $\Theta(n)$, and the server's total communication is $\Theta(n^2)$).

This paper aims to alleviate problem (b) by giving a version that uses a reduced communication graph. The paper argues that if the graph has appropriate properties then the resulting protocol is secure (though that argument is problematic, see below). The paper shows that a random Erdos-Renyi graph with density about $1/\sqrt{n}$ satisfies the properties, leading to a protocol with communication $\tilde O(\sqrt{n})$ per client.

I like the approach and idea of the paper, but I don't think the execution is quite there yet—I don't feel the paper, as written, is acceptable for ICLR.  The paper's value hinges on three claims: correctness (the protocol should complete even when some players drop out), low communication, and security. I did not check the arguments for any of these in detail, but correctness and communication look fine. The main issue is the security argument, which is not properly developed (and not obviously correct). The security model is not clearly articulated, the claims don't clearly describe assumptions on the number of corrupted parties, and so forth. For example, the paper makes claims such as that the protocol prevents membership inference. That can't be true because many (most?) membership attacks use only the final trained model (or even just its outputs)—hiding the intermediate gradients doesn't help.

This paper is about multiparty computation (to be clear, secure summation is a special case of MPC where the ideal functionality is a sum). It should use the language developed over the past thirty+ years in the crypto and security communities to formulate and prove its claims of security. This would allow for clear and refutable claims, and a clear discussion of the new protocols limitations.  In particular, the restriction to semi-honest parties is a huge one, albeit shared with the SA protocol of Bonawitz et al.



* Modeling the exact security properties of protocols with parties that drop out is a bit subtle. See, for example, this paper for discussion: https://eprint.iacr.org/2018/997

* The idea of replacing the complete graph with a low-degree expander to save communication has been used elsewhere. Some relevant citations are below (though there are others, too):

** Fitzi, M., Franklin, M., Garay, J., Vardhan, H.: Towards optimal and efficientperfectly secure message transmission. In: Vadhan, S.P. (ed.) TCC 2007. LNCS,vol. 4392, pp. 311–322. Springer, Heidelberg (2007)

** Harnik, D., Ishai, Y., Kushilevitz, E.: How many oblivious transfers are neededfor secure multiparty computation? In: Menezes, A. (ed.) CRYPTO 2007. LNCS,vol. 4622, pp. 284–302. Springer, Heidelberg (2007)

* Finally: This type of paper might be a better fit for a security or crypto venue, where its contributions can be better evalauted and appreciated. It is up to the authors where to submit, of course, and I don't generally take conference scope too strictly, but the paper isn't really about learning representations, and isn't clearly a good fit for the ICLR audience.

---

> ### Author Response · Authors · 2020-11-17
> **Response to Reviewer 1**
>
> **Need rigorous security model/argument**
> >The security model (or threat model) can be more clearly described as follows: an eavesdropper can access the information transmitted between any client and the server from Step 0 to Step 3 of the CCESA algorithm, namely, public keys, secret shares, masked local models and the indices of survived clients $V_3$. This model is equivalent to the “Server-only” honest-but-curious adversary in the cryptographic language. In page 5 of the revised manuscript, we have clarified our security model. A natural course of development for our work is to extend our privacy results (Theorem 2,4,6) to “Clients-server collusion” adversary by following the steps proving Theorem 6.3 of (Bonawitz, et al., 2017).
>
> **Hiding the intermediate gradients doesn't help to prevent membership inference**
> > We clarify that we consider the federated learning setup where the masked local models (instead of intermediate gradients) are transmitted. In our experiments, we tested the membership inference attack on the trained local model sent from a client to the server.
>
> **Use the language developed in the crypto/security community**
> > Our language was intended for readers familiar with federated learning, not necessarily those well-versed in crypto/security. Our goal was to provide an algorithm that allows the building of secure federated learning systems. Also, our theoretical development is information-theatrically-oriented in the sense that we achieve the maximum entropy of local models conditioned on the information accessible to the eavesdroppers.
>
> **Restriction to semi-honest parties**
> > In page 5 of the revised manuscript, we clearly mentioned that our algorithm is only applicable to the “honest-but-curious (semi-honest)” adversary model.
>
> **Related works on using low-degree graphs**
> > Thank you for the suggestion. We have cited two key previous papers in Sec. 1 of the revised manuscript.

---

### Author Response · Authors · 2020-11-17
**General comments to reviewers**

Thank you for the reviewers’ comments and constructive suggestions. Below we provide answers to the questions and concerns. We believe that our point-by-point responses address all concerns raised.

What we have changed is colored as red in the revised manuscript.

In the revision, the statement of Theorem 2 has been improved. To be specific, in the revised manuscript we present the necessary and sufficient condition on the assignment graph $G$ to satisfy the privacy constraint, while we only provided a sufficient condition in the original manuscript.

---

### Decision · Program_Chairs · 2021-01-07
**Final Decision**

**Decision:**

Reject

**Comment:**

This paper presents an efficient secure aggregation algorithm in federated learning scenarios, which employs sparse random secure-sharing clients. Four experienced reviewers left valuable comments on this paper, and three of them are unfortunately negative to this work (4, 4, 3) while one reviewer is slightly on the positive side.

The reviewers are generally positive about the main idea and the direction for this work, but they are not convinced of its mathematical soundness and practical benefits; the theoretical analysis and mathematical proof has been conducted only for simplified models while their practical advantage is not clear enough. Also, even the most positive reviewer (R3) is concerned about the novelty of the proposed approach.
Although the concerns raised in the original reviews have been partially clarified during the discussion phase, there still remain several critical limitations, which makes this paper require (probably) multiple rounds of revision before publication and this AC has a reservation for accepting this paper.